# Non-covalent ligand-oxide interaction promotes oxygen evolution

Qianbao Wu [1,9], Junwu Liang[2,9], Mengjun Xiao [1,9], Chang Long [1], Lei Li [1], Zhenhua Zeng [3] ✉, Andraž Mavrič [4], Xia Zheng[1], Jing Zhu[5], Hai-Wei Liang [6], Hongfei Liu[1], Matjaz Valant [4], Wei Wang [7], Zhengxing Lv[8], Jiong Li [8] & Chunhua Cui [1] ✉

Strategies to generate high-valence metal species capable of oxidizing water often employ composition and coordination tuning of oxide-based catalysts, where strong covalent interactions with metal sites are crucial. However, it remains unexplored whether a relatively weak "non-bonding" interaction between ligands and oxides can mediate the electronic states of metal sites in oxides. Here we present an unusual non-covalent phenanthroline-$CoO_2$ interaction that substantially elevates the population of $Co^{4+}$ sites for improved water oxidation. We find that phenanthroline only coordinates with $Co^{2+}$ forming soluble $Co(phenanthroline)_2(OH)_2$ complex in alkaline electrolytes, which can be deposited as amorphous $CoO_xH_y$ film containing non-bonding phenanthroline upon oxidation of $Co^{2+}$ to $Co^{3+/4+}$. This in situ deposited catalyst demonstrates a low overpotential of 216 mV at 10 mA cm$^{-2}$ and sustainable activity over 1600 h with Faradaic efficiency above 97%. Density functional theory calculations reveal that the presence of phenanthroline can stabilize $CoO_2$ through the non-covalent interaction and generate polaron-like electronic states at the Co-Co center.

The oxygen evolution reaction (OER) at the anode plays a key role in supplying electron sources for the electrochemical reduction of $H_2O$, $CO_2$, and $N_2$ to fuels and value-added chemicals at the cathode[1]. Current efforts have been devoted to searching for low-cost, highly active, and stable OER catalysts as well as revealing their reaction centers[2–4]. It has been demonstrated that the formation of high-valance metal centers such as $Co^{IV}$, $Fe^{IV}$, and $Ni^{IV}$ under bias is the prerequisite for oxygen evoution[5–9]. Indeed, the polarization-activated 3d transition-metal oxides and (oxy)hydroxides containing concentrated high-valence metal sites are among the most active OER electrocatalysts[9–12].

While the critical role of high-valence metal centers is unraveled, it remains a challenge to deliberately design and construct catalysts that enable the facile generation of these high-valence metal sites under OER conditioning.

To favor the formation of high-valence metal sites, heteroatom Fe[13,14], Cu[15], Co[16], Ni[17], Au[18,19], or Ag[20] have been incorporated into the host metal oxides to lower the thermodynamic barrier[3,8,21–23]. For instance, Fe doping into the $NiO_x$ matrix promotes the formation of $Ni^{4+}$, bringing in improved OER activity[24]. Besides, the combination of Co, Fe, and non-metal P with Ni in the NiCoFeP catalyst leads to

[1]Molecular Electrochemistry Laboratory, Institute of Fundamental and Frontier Sciences, University of Electronic Science and Technology of China, Chengdu 610054, China. [2]Optoelectronic Information Research Center, School of Physics and Telecommunication Engineering, Yulin Normal University, Yulin, Guangxi 537000, China. [3]Davidson School of Chemical Engineering, Purdue University, West Lafayette, IN 47907, USA. [4]Materials Research Laboratory, University of Nova Gorica, Vipavska 13, SI-5000 Nova Gorica, Slovenia. [5]Department of Chemical Physics, School of Chemistry and Materials Science, University of Science and Technology of China, Hefei 230026, China. [6]Hefei National Laboratory for Physical Sciences at the Microscale, Department of Chemistry, University of Science and Technology of China, Hefei 230026, China. [7]School of Materials and Energy, University of Electronic Science and Technology of China, Chengdu 610054, China. [8]Shanghai Synchrotron Radiation Facility, Shanghai Advanced Research Institute, Chinese Academy of Sciences, Shanghai 201210, China. [9]These authors contributed equally: Qianbao Wu, Junwu Liang, Mengjun Xiao. ✉e-mail: zeng46@purdue.edu; chunhua.cui@uestc.edu.cn

abundant $Ni^{4+}$ sites[11], thereby improving OER performance as well. Despite major progress in valence-relevant enhancements of OER activity, the structural and compositional complexity of multi-metal oxides has obstructed the exact mechanism studies, as a consequence, the precise active metal centers are not identified[9,25]. This is because some species in the electrochemically amorphized surface layers may be gradually converted into soluble oxygenated metal anions[26–29], such as Fe anions ($FeO_4^{2-}$)[30,31]. As a result, the concomitant surface reconstruction can change the local composition, coordination environment, and electronic structure.

Catalysts containing single-metal components generally do not have the above issues faced by multi-metal oxides, and thus can serve as well-defined candidates for mechanism study. For instance, the high-oxidation state $Co^{4+}$ sites in CoPi[32], $Co_4O_4$ cubane[33,34], and non-heme $Co^{4+}$-O complex[35] have been revealed as the key intermediates for the formation of O-O bond[5,36–38]. To stabilize $Co^{4+}$ under the OER process, the Co sites were usually immobilized in a matrix or ligand network, such as substrate-supported single-atom catalysts with strong covalent interactions[39–41]. It is evidenced that those covalent supports play an important role in facilitating the formation of $Co^{4+}$. Whereas, these structurally well-defined catalysts tend to decompose along with a decreased concentration of $Co^{4+}$ under harsh electrochemical conditions[42–45]. Thus, clarifying their delicate valence-activity relationships and further increasing OER activity remains a challenge.

In this study, we report the discovery of a non-covalent ligand-metal oxide interaction that allows the formation of abundant $Co^{4+}$ sites to enhance OER performance. In contrast to the strong bonding interaction in the compounds, this non-covalent interaction reported here features a relatively weak yet crucial interaction between metal sites and chelating ligands. Both in situ and ex situ measurements together with DFT calculations suggest that the non-covalent interaction enables the facile transition from $Co^{3+}$ to $Co^{4+}$. The computational free energy diagrams and experimental evidence jointly reveal the valence-dependent coordination "switch" between soluble covalent $Co(phenanthroline)_2(OH)_2$ complex, abbreviated as $Co(phen)_2(OH)_2$, in solution and non-covalent ligand-modified $CoO_xH_y$ on electrodes (labeled as Co-PH with high content of $Co^{4+}$). This feature suggests that the active $Co^{4+}$ sites can regenerate, with $Co(phen)_2(OH)_2$ complex in the electrolytes. Consequently, a self-optimized low overpotential of 216 mV was reached at 10 mA cm$^{-2}$ for over 1600 h on the Co-PH catalyst.

Based on DFT calculations, we demonstrate that the $CoO_2$/phen hybrid outperforms bulk $CoO_2$, ascribing to the increase of density of

states near the Fermi level and electron charge transfer between the components. The non-covalent interaction leads to polarons in the $Co^{4+}$-enriched structure, which facilitates the deprotonation of the surface bridge $OH^*$ species and subsequently the formation of O-bridged dual Co-Co moieties. Moreover, the calculated charge density differences and magnetic moments show how $CoO_xH_y$ and phen work in synergy with the phen-induced polarons to reduce the thermodynamic overpotential from 0.75 to 0.4 V. This non-covalent interaction provides a novel pathway to tune the catalytic properties of heterogeneous catalysts.

## Results
### Discovery of ligand-facilitated $Co^{4+}$ formation

The chronoamperometry (CA) was performed on pristine CoOOH model catalysts in both phen-free and phen-containing 1.0 M NaOH electrolytes at 1.7 V versus a reversible hydrogen electrode ($V_{RHE}$). As shown in Supplementary Fig. 1, the presence of phen in the electrolyte leads to a continuous increase in OER current density from 6.4 to 11.7 mA cm$^{-2}$ during the 10 h chronoamperometry test. In sharp contrast, the current density decreases from 6.4 to 4.0 mA cm$^{-2}$ in the absence of phen. We further recorded the OER polarization curves after CA tests (Fig. 1a). The overpotential at 10 mA cm$^{-2}$ reduces by 80 mV for the CoOOH treated in the phen-containing electrolyte (abbreviated as pc-CoOOH) compared to the pristine CoOOH, while that is increased by about 20 mV for phen-free electrolyte treated CoOOH (pf-CoOOH). Both the CA and polarization tests illustrate that the introduction of phen into the alkaline electrolyte can improve the OER activity of CoOOH. It is worth noting that the electrochemical active surface area (ECSA) of CoOOH after 10 h treatment in phen-containing NaOH remains almost unchanged (Supplementary Fig. 2). This excludes the activity enhancement from the variation of surface area. We further quantified the Co ions concentration in electrolytes after the chronoamperometry test. The leached Co in the phen-free NaOH reached up to 660 µg L$^{-1}$, which is sixfold higher than that in the phen-containing NaOH (Fig. 1b and Supplementary Fig. 3), indicating that phen ligand relieves Co dissolution during OER.

To understand the enhanced OER activity as well as the reduced dissolution of Co in the presence of phen in the electrolyte, we investigated the Co chemical state and coordination environment of as-prepared pc-CoOOH and pf-CoOOH by X-ray absorption fine structure spectroscopy (XAFS). The Co K-edge X-ray absorption near edge structure spectra (XANES) of pc-CoOOH shifts toward higher energy relative to that of pf-CoOOH, indicating a higher Co oxidation

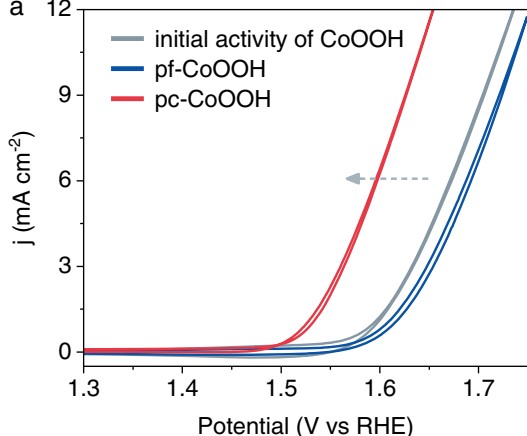
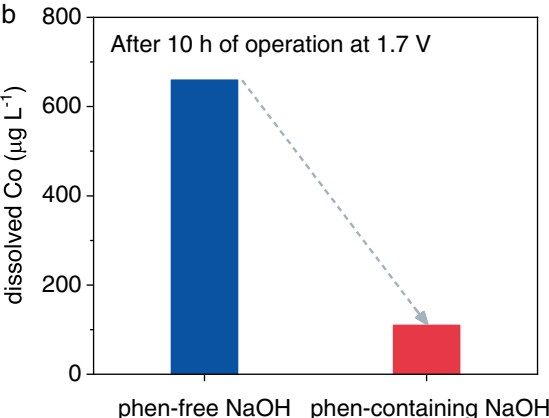

**Fig. 1 | Ligand-promoted oxygen evolution activity. a** The *j-V* curves of pf-CoOOH and pc-CoOOH. "pc" and "pf" represent that the pristine CoOOH precatalysts were treated at 1.7 $V_{RHE}$ for 10 h in phen-containing and phen-free 1.0 M NaOH electrolytes, respectively. **b** The concentration of Co leached into electrolytes (measured by ICP-MS after 10 h chronoamperometry at 1.7 $V_{RHE}$).

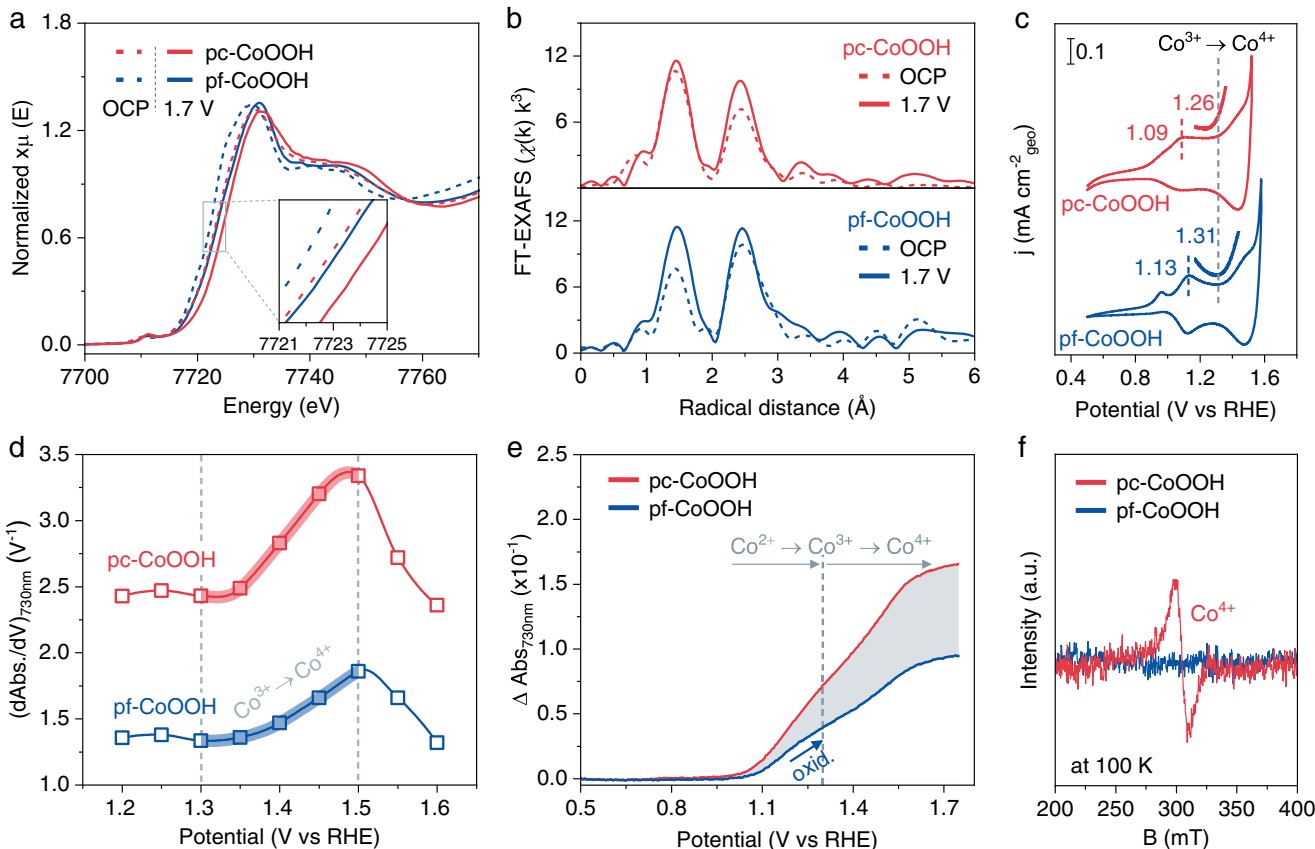

**Fig. 2 | Ligand facilitated the formation of concentrated Co⁴⁺. a** Co K-edge XANES of pf-CoOOH and pc-CoOOH at OCP and 1.7 $V_{RHE}$, and **b** the corresponding k³-weighted Fourier-transformed EXAFS. **c** The redox couples of pf-CoOOH and pc-CoOOH in 1.0 M NaOH, the enlarged insets represent the onset oxidation potential from Co³⁺ to Co⁴⁺. **d** The difference in differential absorbance at 730 nm for both pc-CoOOH and pf-CoOOH in 1.0 M NaOH as a function of applied potential. **e** The potential-dependent UV-Vis spectra at 730 nm. The background was deducted at 0.5 $V_{RHE}$. **f** The ex situ EPR of pf-CoOOH and pc-CoOOH at 100 K.

state in pc-CoOOH (Supplementary Fig. 4a)[38,41]. The k³-weighted Fourier-transformed Co K-edge extended X-ray absorption fine structure (EXAFS) reveals a Co-O distance of 1.93 Å and a Co-Co distance of 2.89 Å[46,47]. While the Co-O and Co-Co distances of pc-CoOOH and pf-CoOOH keep the same, pc-CoOOH exhibited a relatively lower Co-Co coordination number (Supplementary Fig. 4b–d). This suggests a more pronounced fragmentation of the Co-O-Co network for pc-CoOOH[48,49], in line with the aggravated amorphization (cf. the selected area electron diffraction (SAED) and SEM images in Supplementary Figs. 5–7). The amorphization together with a low concentration of Co ions in the electrolyte for pc-CoOOH suggests the presence of phen ligand accelerates the Co redeposition.

We performed in situ XAFS measurements to understand the variations of catalysts during the OER. As is confirmed by a shift (1.22 eV) of XANES to higher absorption energy upon the potential switching from open circuit potential (OCP) to 1.7 $V_{RHE}$ for pc-CoOOH than that of pf-CoOOH (Fig. 2a and Supplementary Figs. 8, 9; the images of the in situ XAFS equipment and the electrochemical cell were shown in Supplementary Fig. 8e, f), a higher Co valence state about +3.9 at 1.7 $V_{RHE}$ in pc-CoOOH was concomitantly generated during the OER process. In addition, relative to pf-CoOOH, the Co-O coordination number of pc-CoOOH increased from 5.11 at OCP to 6.05 at 1.7 $V_{RHE}$[50] (Fig. 2b, Supplementary Figs. 10, 11, and Supplementary Table 1), and the increased Co-Co coordination number for pc-CoOOH is likely due to the more favorable formation of di-μ-oxo or μ-hydroxo bridged Co species[46].

To further clarify the effect of phen on OER, we systematically studied the redox behavior of pc-CoOOH and pf-CoOOH. As shown in Fig. 2c, the Co²⁺/Co³⁺ oxidation peak of pc-CoOOH negatively shifts by

~40 mV relative to that of pf-CoOOH. Further, compared to pf-CoOOH, the onset potential for the oxidation of Co³⁺ to Co⁴⁺ on pc-CoOOH negatively shifts from 1.31 to 1.26 $V_{RHE}$. We also applied in situ UV-Vis[46] to track the Co⁴⁺ formation kinetics as a function of applied bias (Supplementary Fig. 12). The much steeper differential absorbance between 1.3 and 1.5 $V_{RHE}$ for pc-CoOOH illustrates the faster generation of Co⁴⁺ in the presence of phen (Fig. 2d)[51,52]. Importantly, pc-CoOOH presents ~1.8 times higher absorbance intensity for Co⁴⁺ (at 730 nm) relative to pf-CoOOH at 1.75 $V_{RHE}$ (Fig. 2e), indicating an easier charge accumulation on Co sites[22], thus a much higher population of Co⁴⁺. Besides, we studied the chemical states of both pf-CoOOH and pc-CoOOH catalysts by ex situ X-ray photoelectron spectroscopy (XPS) and electron paramagnetic resonance (EPR). Co 2p XPS spectra were fitted according to the reported binding energies of Co²⁺, Co³⁺, and Co⁴⁺[53–55], which suggested an increased Co³⁺/Co²⁺ ratio from 1.12 to 1.32 as well as the appearance of Co⁴⁺ after introducing phen ligand (Supplementary Fig. 13 and Supplementary Table 2). Since Co⁴⁺ is an EPR-active species (with g-values ranging from 2.1 to 2.4 depending on the coordination environments)[56–59], we used EPR spectroscopy to further quantify it. The spectrum of pc-CoOOH displays a relatively strong signal centered at g ≈ 2.20 which can be attributed to a low-spin (S = 1/2) Co⁴⁺. In contrast, there is only a very weak signal shown at the same g value for the pf-CoOOH catalyst. This again indicates the presence of phen benefits Co⁴⁺ formation (Fig. 2f).

## Modeling the interplay between Co⁴⁺ and phen

To reveal the crucial role of phen ligand in the Co valence transitions under anodic polarization, we studied the phen-embedded layered CoOOH (represents the chemical state with 100% Co³⁺) and CoO₂

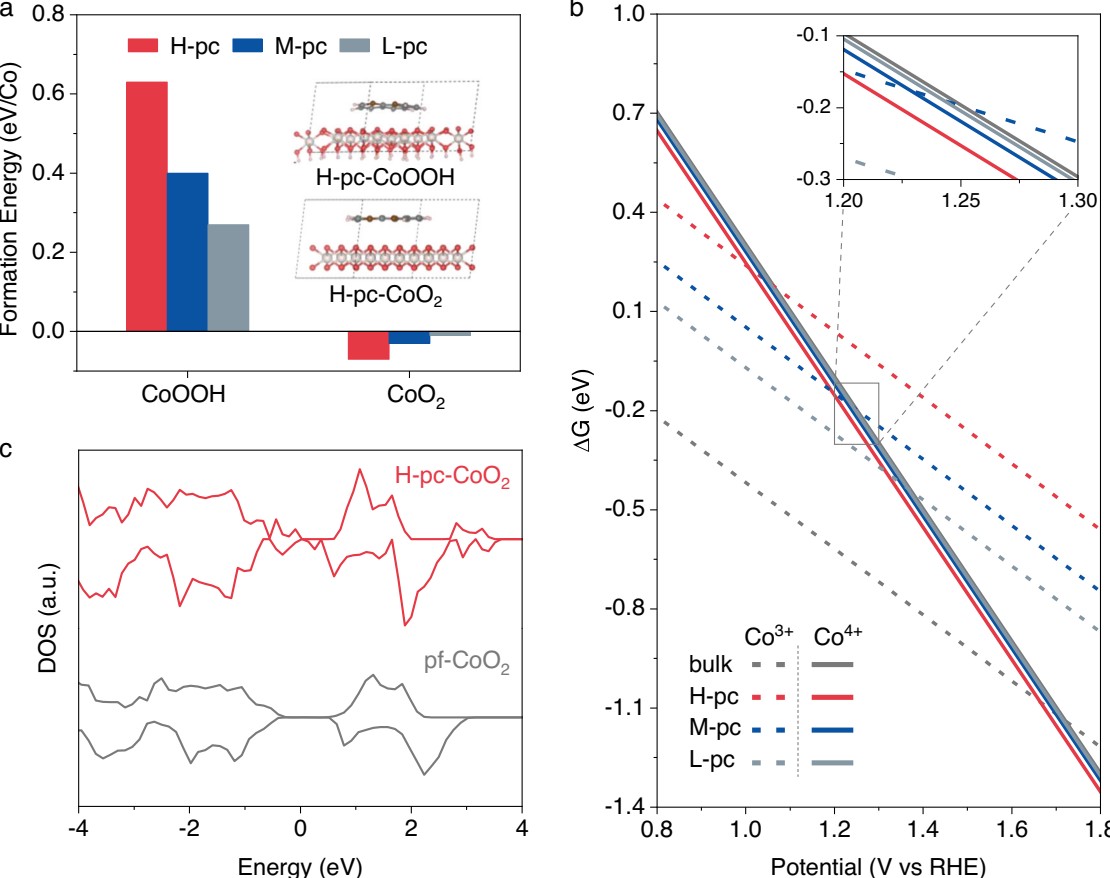

**Fig. 3 | DFT calculations. a** The formation energy of layered CoOOH and CoO$_2$ containing high(H)-, middle(M)-, and low(L)-density of phen (i.e., 1, 0.5, and 0.33 ML phen) in the interlayers. Inset shows the models of high-density of phen inserted into layered CoOOH and CoO$_2$. Metal Co and C, N, O, and H are represented by yellowish, gray, brown, red, and pink balls, respectively. **b** The free energy diagrams of phen-free and H-, M-, and L- density of phen-inserted layered CoOOH and CoO$_2$. **c** The densities of states (DOS) of phen-free and H-density of phen-inserted CoO$_2$.

(100% Co$^{4+}$) by density functional theory (DFT) calculations (Supplementary Figs. 14–17). As shown in Fig. 3a and Supplementary Table 4, the intercalation of phen into CoOOH is endothermic, with the exact magnitude depending on the content of phen, which indicates that the interaction between phen and CoOOH is thermodynamically unfavorable. Conversely, the intercalation of phen into CoO$_2$ is exothermic. As a consequence, the presence of phen leads to a thermodynamically more favorable transition from CoOOH to phen-intercalated CoO$_2$. Additionally, the transition potential decreases with the increase of phen content, i.e., from 1.72 V$_{RHE}$ for the bulk phase to 1.37 V$_{RHE}$, 1.23 V$_{RHE}$, and 1.01 V$_{RHE}$ for 1/3 ML (ML: monolayer), 2/3 ML, and 1 ML of phen covered CoOOH, respectively (Fig. 3b). The intercalation of phen results in the exfoliation of CoO$_2$ sheets from bulk CoOOH, which explains phen-induced amorphization of CoOOH at 1.7 V$_{RHE}$ (Supplementary Fig. 5).

Notably, bulk CoO$_2$ is a semiconductor with high resistance, which is unfavorable for charge transfer, thus greatly restricting its OER activity. However, phen intercalation increases the density of states near the Fermi level (Fig. 3c and Supplementary Fig. 18) and thereby improves the charge transfer kinetics. This is because of the formation of a polaron-like Co$^{3+}$ site within the enhanced content of Co$^{4+}$ (Supplementary Fig. 19)[60–64]. This unique structure plays a central role in the improved OER activity, as discussed further below.

**Valence-dependent interactions between Co and phen**

The theoretical study suggests that the inclusion of phen into the interlayer of CoOOH is thermodynamically unfavorable

(Supplementary Table 4). Thus, the concentration of phen in pc-CoOOH should be very low. To further clarify the interplay between phen and Co in different oxidation states, we prepared Co(OH)$_2$ and CoOOH model catalysts and studied their coordination ability with phen. Interestingly, after introducing phen into 1.0 M NaOH, the Co(OH)$_2$ film rapidly dissolved into the electrolyte within 10 min at OCP. However, CoOOH remained unchanged on electrodes under the same conditions (Supplementary Fig. 21). Hence, we reasoned that phen tends to coordinate with paramagnetic Co$^{2+}$ instead of diamagnetic Co$^{3+}$ to form a soluble complex. To further verify it, the CoOOH electrode was held at 0.7 V$_{RHE}$ in phen-containing 1.0 M NaOH to allow the reduction of Co$^{3+}$ to Co$^{2+}$ (Supplementary Fig. 22). As expected, we observed a complete dissolution of the catalyst film, which led to the disappearance of OER activity (Supplementary Fig. 23). Combing all the experimental and theoretical results together, we conclude that phen has three types of interrelations with Co: (i) phen coordinates with Co$^{2+}$ to form a soluble complex in 1.0 M NaOH; (ii) phen presents a thermodynamically unfavorable interplay with Co$^{3+}$; iii) phen facilitates the conversion of Co$^{3+}$ to Co$^{4+}$, which can be stabilized on electrodes through non-covalent interaction under OER conditioning (Fig. 4a and Supplementary discussion to Supplementary Figs. 20–23).

Regarding the phen-assisted Co valence state transition, we further analyzed the chemical states and coordination environments of Co according to the ΔG−potential diagram. Co(phen)$_2$(OH)$_2$ molecular model (vide infra) was used as a representative soluble complex. We found that, as expected, Co(phen)$_2$(OH)$_2$ converted to the phen-attached monolayered CoO$_2$ at 1.37 V$_{RHE}$, rather than the soluble

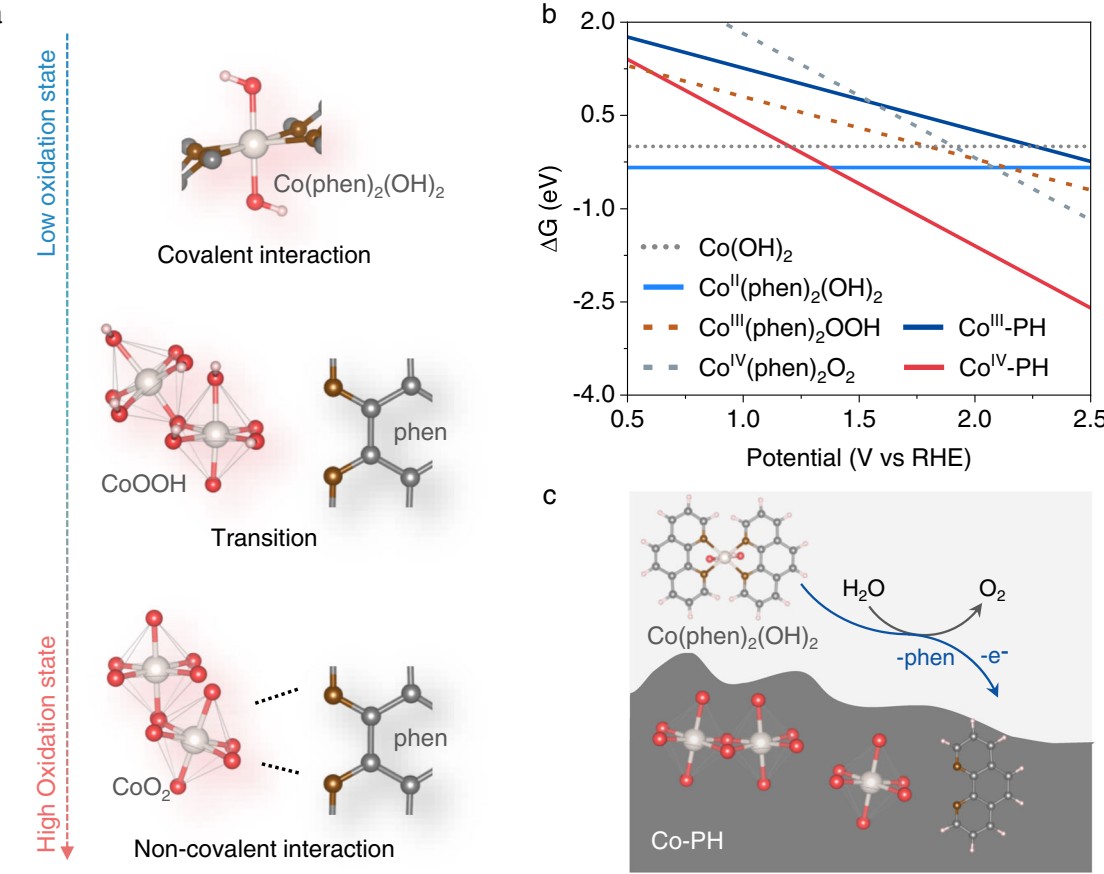

**Fig. 4 | Valence-dependent interactions between Co and phen. a** The scheme of valence-dependent interactions between Co and phen. **b** The free energy diagrams of $Co(OH)_2$, $Co^{II}(phen)_2(OH)_2$, $Co^{III}(phen)_2OOH$, $Co^{IV}(phen)_2O_2$, phen-inserted CoOOH ($Co^{III}$-PH), and phen-inserted $CoO_2$ ($Co^{IV}$-PH). **c** The scheme for the electrodeposition of phen-inserted high-valence $CoO_xH_y$ fragments from soluble $Co(phen)_2(OH)_2$ species. The yellowish, gray, brown, red, and pink balls represent metal Co and C, N, O, and H, respectively.

$Co(phen)_2O_2$ or $Co(phen)O_2$ molecules whose transition potentials were over 2.0 $V_{RHE}$ (Fig. 4b and Supplementary Figs. 24, 25). Thus, electrochemical oxidation of soluble $Co(phen)_2(OH)_2$ leads to the formation of phen-embedded high-valence state $CoO_xH_y$ fragments (Fig. 4c) instead of oxidized complexes with covalent bonds[65–68].

**In situ deposition of $CoO_xH_y$ containing non-bonding phen**
Based on the above findings, we employed electrodeposition to prepare the amorphous $CoO_xH_y$ film containing non-bonding phen (Co-PH) directly from soluble Co-phen complex in 1.0 M NaOH. The soluble Co-phen electrolyte was prepared by mixing $Co^{2+}$ salts with phen in deionized water, whose pH was further adjusted to 13.9 by NaOH at 298 K. The final concentration of NaOH is about 1.0 M. UV-Vis in combination with EPR[56,69] suggest a hydroxyl-involved Co-N coordination state (Supplementary discussion to Supplementary Figs. 26–28). $^{18}$O-labeled and D-labeled Fourier transform ion cyclotron resonance mass spectrometry (FT-ICR-MS) confirms that the molecular formula is $Co(phen)_2(OH)_2$, which is consistent with the prediction by DFT calculations (Supplementary discussion to Supplementary Figs. 29, 30).

The Co-PH films were electrodeposited from soluble $Co(phen)_2(OH)_2$ in 1.0 M NaOH (Supplementary discussion to Supplementary Figs. 31–38). In situ EPR was used to track the $Co^{4+}$ in the oxide film after electrodeposited at 1.7 $V_{RHE}$ for different time scales (Fig. 5a). A paramagnetic $Co^{4+}$ signal[5] (S = 1/2) located at g ≈ 2.25 was captured at room temperature (298 K). This $Co^{4+}$ signal became more pronounced when the temperature was cooled down to 100 K (Fig. 5b). In situ UV-Vis spectra revealed the accumulation of Co-PH on the FTO electrode, as was confirmed by the gradually enhanced absorbance between 350 and 550 nm (Supplementary Fig. 39). Furthermore, the mass of electrodeposited Co-PH catalysts was quantified by electrochemical quartz crystal microbalance (EQCM). It exhibited a linear relationship with the OER current density, suggesting a close correlation between the amount of Co sites and the OER activity. (Fig. 5c). The intrinsic activity of freshly-prepared Co-PH was evaluated based on the mass activity and turnover frequency (TOF) (Fig. 6a and Supplementary Fig. 39). The mass activity of electrodeposited Co-PH displays ~100 times (1.67 A $mg^{-1}$ at an overpotential of 350 mV) enhancement in comparison to that of pf-CoOOH. In addition, Co-PH was found to have a high TOF value of 0.35 $O_2$ $s^{-1}$ per total metal site under 350 mV overpotential, which is ~140 times higher than that of pf-CoOOH. This even outperforms the typical Co-based layered double hydroxides and (oxy) hydroxides (Supplementary Table 6).

**Activity and stability of in situ deposited Co-PH catalyst films**
According to the Co-PH mass-OER activity relationship (Fig. 5c), a continuous deposition at 1.7 $V_{RHE}$ would further increase the OER activity to the optimal state. The deposited Co-PH catalyst film is visually homogeneous and transparent with a nanostructured texture (Supplementary Figs. 40–42). Interestingly, the in situ deposited Co-PH after 10 min exhibited a Co/N ratio of 1.96 (Supplementary Table 7), which is much lower than that of pc-CoOOH (~8.49, cf. Supplementary Table 2) after 10 h at 1.7 $V_{RHE}$, suggesting much higher density of phen in the in situ electrodeposited Co-PH film. In addition, the deposited Co-PH catalysts after 10 min and 10 h show almost the same Co/N ratio of ~2.0 in the films (Supplementary Fig. 43 and Supplementary

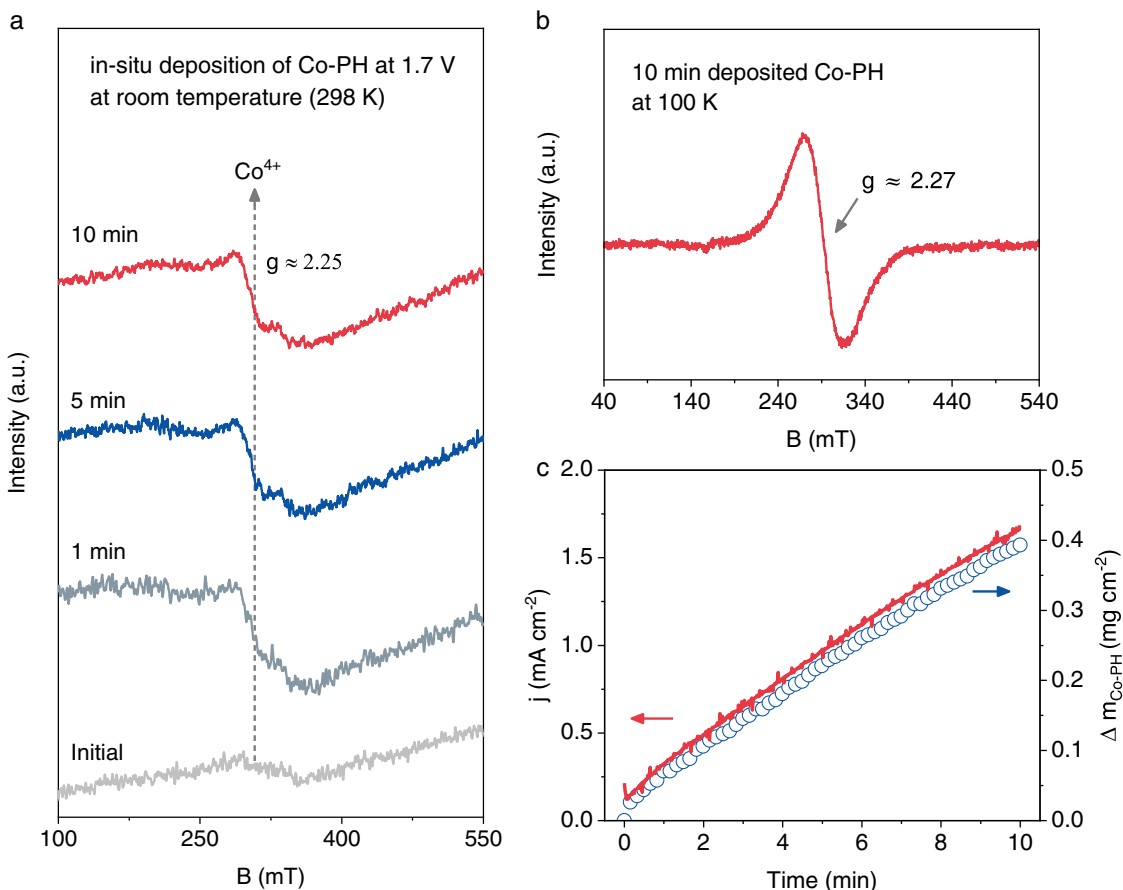

**Fig. 5 | In situ electrodeposition of high-valence Co-PH film containing non-bonding phen. a** In situ EPR at 298 K of the electrodeposited Co-PH catalyst at different time scales. **b** The ex situ EPR of the Co-PH catalyst at 100 K after 10 min-deposition. **c** The relationship between OER current density at 1.7 $V_{RHE}$ and mass (measured by EQCM) of the electrodeposited Co-PH catalyst.

Table 7), exhibiting a comparable amount of phen during the film deposition. Compositional characterizations of the catalyst films further verified the presence of phen after 40 and 500 h electrocatalysis at 10 mA cm$^{-2}$ (Supplementary Figs. 44–46). Moreover, ex situ UV-Vis and attenuated total reflection-Fourier transform infrared (ATR-FTIR) also revealed the existence of phen in the deposited Co-PH (Supplementary Figs. 47, 48). Importantly, the phen was observed intact in Co-PH films after OER operation (Supplementary Figs. 49, 50). Deliberate decomposition of phen in Co-PH film by irradiation with 254 nm laser[36] led to the decay of OER performance, again highlighting the critical role of phen ligand in enhancing the OER activity (Supplementary Fig. 51 and Supplementary Table 8).

We further evaluated the long-term stability of the Co-PH catalyst at 10 mA cm$^{-2}$. To better understand the whole process and activity variation trend of Co-PH, we present both the catalyst deposition process and the subsequent OER stability test in one curve in Fig. 6b. The overpotential decreases continuously with the extended operation. The catalyst deposition and the subsequent self-optimizing process at 10 mA cm$^{-2}$ are relatively slow and gradually tend to the best state, and this process could be accelerated by increasing the current densities (Supplementary Fig. 52). As shown in Fig. 6b, this catalyst achieved a lifetime of more than 1600 h (~68 days) at 10 mA cm$^{-2}_{geo}$. In addition, online gas chromatography (GC) analysis further indicated a Faradic efficiency of >97% for O$_2$ at 10 mA cm$^{-2}$ (Supplementary Fig. 53). Remarkably, regardless of whether the current density was normalized to the geometric area or electrochemically active surface area (ECSA), Co-PH features a low overpotential of 216 mV at 10 mA cm$^{-2}$, about 330 mV lower than that

of CoOOH catalysts measured in the current work and the literature (Fig. 6b and Supplementary Figs. 54, 55). Notably, the phen is stable in the catalyst even after ~1600 h operation (Fig. 6b, inset). Benefiting from the phen-endowed self-healing ability, the lifespan of Co-PH could be much larger than that. These performances place unary Co-PH among the most active and durable OER catalysts (Fig. 6c, d and Supplementary Table 9).

**Understanding improved activity through modeling**

To better understand the role of phen, we further studied OER activity through DFT calculations. We first evaluated the steady-state configurations of CoO$_2$ (10-10) and Co-PH (10-10) surfaces (Fig. 7a). The calculated surface free energy diagrams of CoO$_2$ and Co-PH indicate that, under OER conditions, coordinatively unsaturated surface O sites are saturated with H$_{ad}$ by forming bridge OH$^*$ species, and coordinatively unsaturated Co sites are saturated with OH$_{ad}$ by forming atop OH$^*$ (Fig. 7b and Supplementary Fig. 56). Thus, OER starts from the deprotonation of the surface OH$^*$, as has been proposed in the previous reports[16,23,70]. Also, we found that OH$^*$ deprotonation to O$^*$ is the potential-determining step (PDS) for both bridge OH$^*$ and atop OH$^*$ (Fig. 7d and Supplementary Fig. 57). For the deprotonation of bridge OH$^*$, the theoretical overpotential is about 0.2-0.6 V lower than that of atop OH$^*$ (Supplementary Fig. 58). Thus, OH bridged dual Co-Co sites are reaction centers.

For CoO$_2$, the theoretical overpotential is 0.75 V (Fig. 7d). The presence of polaron-like Co$^{3+}$ induced by phen, however, can reduce the thermodynamic overpotential to 0.4-0.5 V, which is dependent on the relative distance of the reaction center to the location of Co$^{3+}$ site

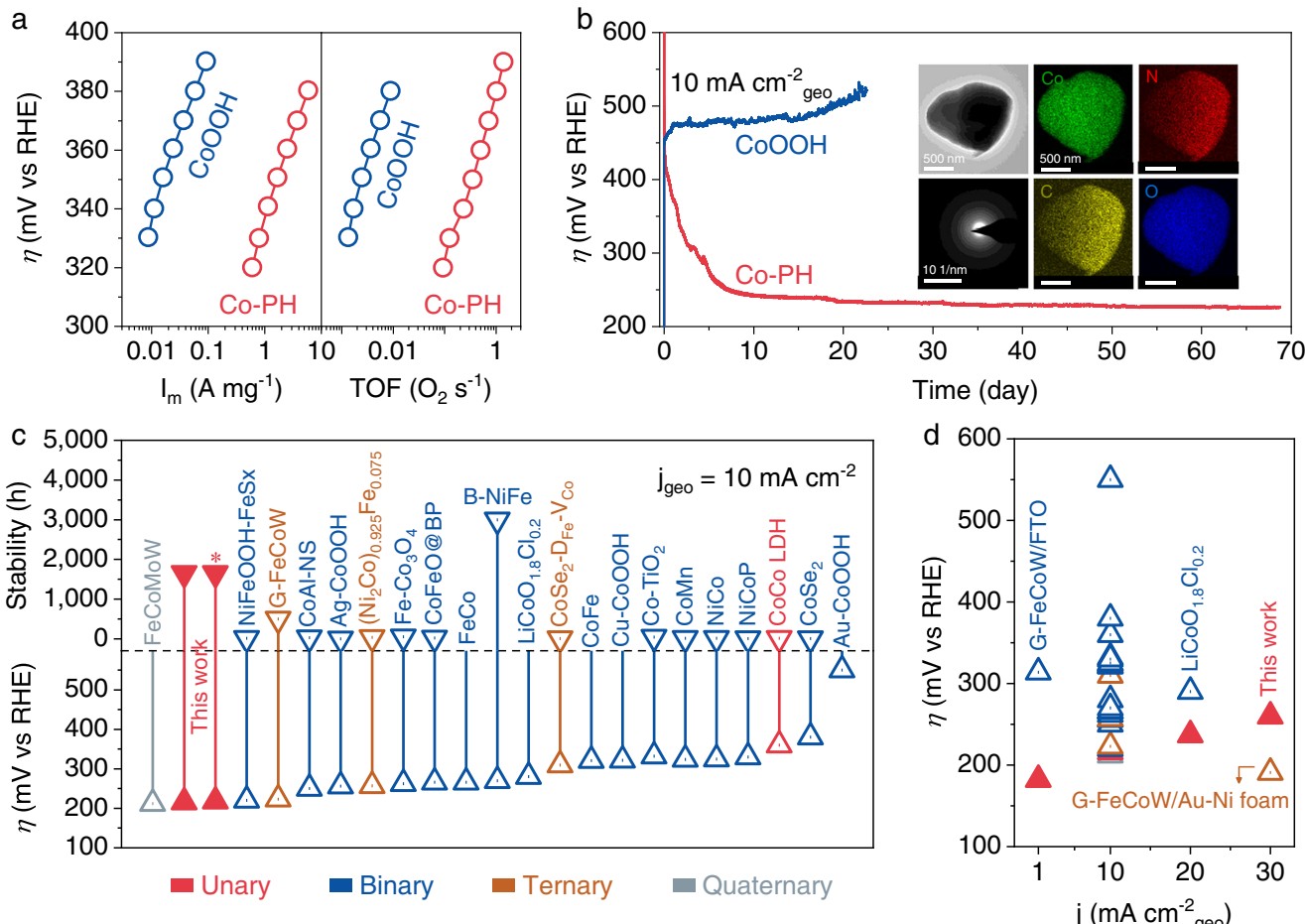

**Fig. 6 | Phen-enriched Co⁴⁺ for efficient water oxidation. a** Tafel plots of the OER mass activity and TOF of as-prepared CoOOH and electrodeposited Co-PH. Co-PH was freshly deposited in $Co(phen)_2(OH)_2$-containing 1.0 M NaOH for 10 min at 1.7 $V_{RHE}$. **b** Stability tests of Co-PH and CoOOH at 10 mA cm⁻² on planar FTO. The stability test of the as-prepared CoOOH was operated in 1.0 M NaOH. Co-PH was in situ deposited on bare FTO and its stability test was operated in $Co(phen)_2(OH)_2$-containing 1.0 M NaOH. Insets: the TEM, SAED, and corresponding element mapping (EDX) images of Co-PH after the stability test. **c** The activity and stability comparison of multi-metal Co-based catalysts with our single-metal Co-PH catalyst at 10 mA cm⁻²$_{geo}$. The overpotential of Co-PH at 10 mA cm⁻² was obtained from the stability test in Fig. 6b. *Noting that the overpotential and stability were obtained at 10 mA cm⁻²$_{ECSA}$. **d** The activity comparison of the above catalysts at different current densities (more details in Supplementary Table 9).

(Fig. 7c, d and Supplementary Figs. 59–69). The reaction center with the lowest thermodynamic overpotential of 0.4 V is the one with Co³⁺ forming the Co³⁺-Co⁴⁺ reaction center. This type of reaction center facilitates the potential-determining deprotonation of OH* to O* which is accompanied by the oxidation of Co³⁺ to Co⁴⁺, as observed by the change of Co magnetic moment from 0 to 1 μB (Supplementary Figs. 62–65). However, even if the Co³⁺ site is not part of the reaction center, i.e., forming Co⁴⁺-Co⁴⁺ type reaction centers, the thermodynamic overpotential only increases slightly (Supplementary Figs. 66–69). For example, for the cases of Co³⁺ with one atom away from the reaction center, the thermodynamic overpotential only increases 0.02–0.05 V due to the effect of polaron, i.e., Co³⁺ is still oxidized to Co⁴⁺ during the deprotonation of OH* to O* (Supplementary Figs. 62–65). The feature of polaron is more clearly illustrated in the cases where Co³⁺ is far away from the reaction center, e.g., on the opposite side of the slab model. For these cases, the thermodynamic overpotential is only up to 0.1 V higher (Supplementary Figs. 66–69). These results highlight the importance of non-covalent ligand-oxide interactions in reducing the OER overpotential by generating abundant Co⁴⁺, active polarons, and dynamic reaction centers.

This work presents a ligand strategy to promote the formation of OER-required high-oxidation state Co⁴⁺ species along its synergy with polaron-like Co³⁺ species through a non-covalent interaction between homogeneous phenanthroline ligand and heterogeneous oxide. Multiple in situ, ex situ characterizations and theoretical computation indicate phenanthroline ligands can be favorably embedded into $CoO_xH_y$ through dissolution/redeposition-induced reconstruction. This process facilitates the formation of highly oxidized $CoO_2$ fragments under anodic polarization. Moreover, we further demonstrate in situ deposition of amorphous $CoO_xH_y$ film containing non-bonding phenanthroline with a high content of Co⁴⁺ directly from soluble $Co^{2+}(phen)_2(OH)_2$ complex in alkaline electrolytes. This unusual ligand-oxide interaction not only substantially elevates the content of active Co⁴⁺ sites but also leads to the formation of polaron-like Co³⁺ sites within abundant Co⁴⁺ species, which improves OER performance through either local or non-local Co³⁺-Co⁴⁺ synergy, resulting in an overpotential of 216 mV at 10 mA cm⁻² over 2 months. This work highlights the non-covalent interaction between heterogeneous catalysts and chelating ligands as a new pathway to optimize electrocatalytic activity and durability.

## Methods

### Chemicals

High-purity CoCl₂·6H₂O, CoCl₂, CoSO₄·7H₂O, and 1,10-phenanthroline were purchased from Adamas without further purification. The high-purity sodium hydroxide was purchased from Sigma-Aldrich and was

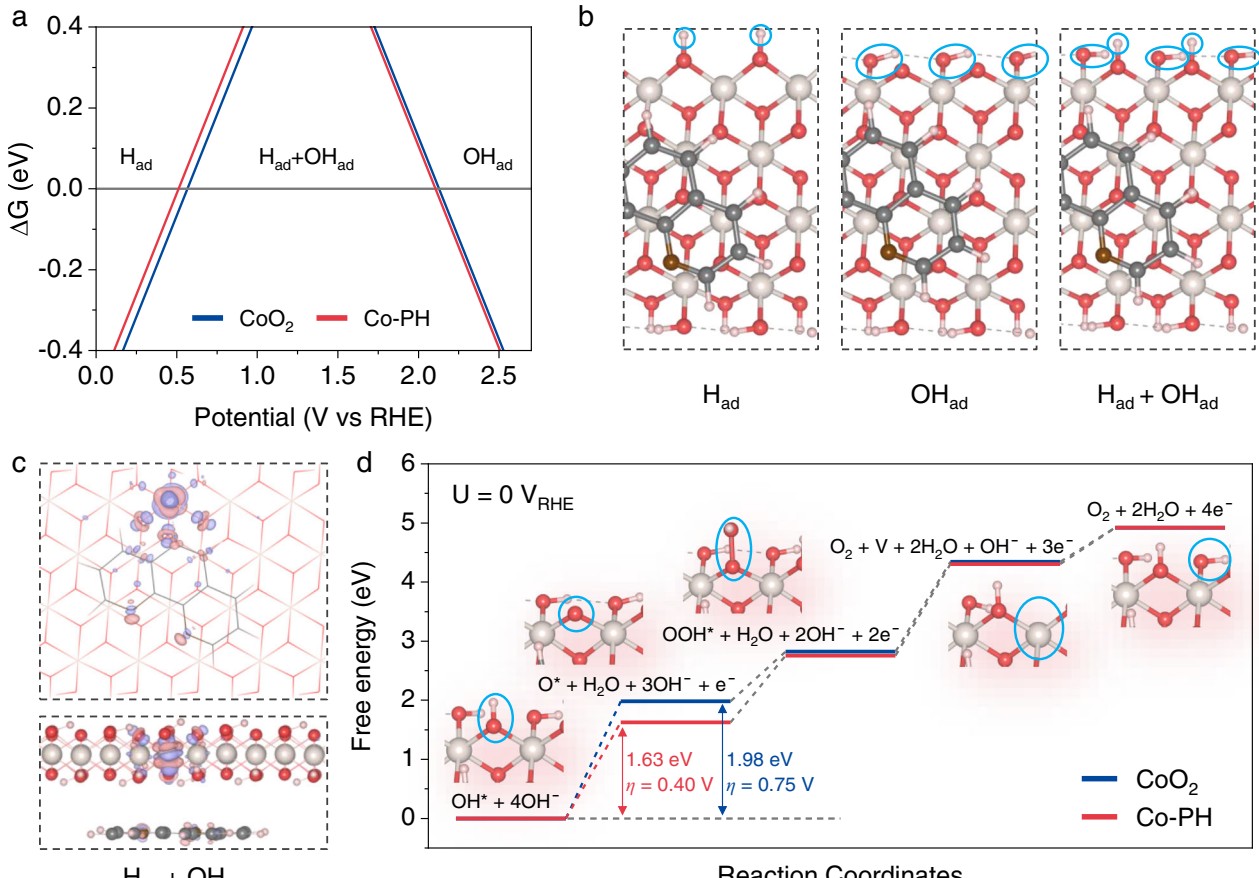

**Fig. 7 | OER mechanism. a** Surface free-energy diagram of $CoO_2$ and Co-PH respective to the surface saturated with hydrogen adsorption ($H_{ad}$) on bridge O ($O_{bri}$) and OH adsorption ($OH_{ad}$) on coordinatively unsaturated sites. **b** The structures of different surface phases on Co-PH, and adsorbates of surface phases are highlighted by light blue circles on the top views. **c** The top view (upper) and side view (under) of charge density differences of Co-PH that surface saturated with $H_{ad}$ and $OH_{ad}$, that is, $H_{ad} + OH_{ad}$ ($OH^*$). The purple isosurfaces represent an increase in the total charge density, and the orange isosurfaces represent a decrease in the total charge density. **d** The reaction-free-energy diagrams for OER on $CoO_2$ and Co-PH at $0$ $V_{RHE}$. Insets are the corresponding enlarged models. The yellowish, gray, brown, red, and pink balls represent metal Co and C, N, O, and H, respectively.

further purified to remove the iron impurities. Fluorine-doped tin oxide glass (FTO) with resistance <15 ohm/sq was purchased from Kaivo FTO-P003. $H_2^{18}O$ ($^{18}O$ abundance 97.4%) and $D_2O$ (D abundance 99.9%) were purchased from Meryer. The $Li^{18}OH$ was prepared by directly dissolving the lithium tablets (>99.99%) in $H_2^{18}O$ in an ice bath condition, and the obtained $Li^{18}OH$ solution was evaporated in a vacuum oven at 45 °C to obtain the dry $Li^{18}OH$ powder. All isotope chemicals were used without further dilution. All the solutions without isotopes were prepared using ultrapure deionized water (measured resistivity 18.2 MΩ cm$^{-1}$ at 25 °C, Milli-Q).

**In situ deposition of Co-PH**
The Co-phen solutions were prepared by adding 4 mM $CoCl_2$·$6H_2O$, 14 mM phenanthroline into 40 mL deionized $H_2O$, and thereafter mixed with 2.0 M NaOH (v/v = 1:1) to form $Co(phen)_2(OH)_2$ containing electrolyte. Typically, the Co-PH catalyst film was in situ deposited on FTO at a constant potential of 1.7 $V_{RHE}$ in the $Co(phen)_2(OH)_2$ containing 1.0 M NaOH electrolyte at room temperature (298 K). Besides, the counter Pt electrode was separated by a Nafion membrane and was immersed in a 1.0 M NaOH electrolyte as well.

**Synthesis of $Co(OH)_2$ and CoOOH**
Typically, the $Co(OH)_2$ on FTO was prepared by referring to a previous report[71]. Specifically, 0.233 g $Co(NO_3)_2$· $6H_2O$, 0.07 g of $NH_4F$, and 0.240 g of $CO(NH_2)_2$ were dissolved in 100 mL deionized water under

stirring for 30 min. Then, the obtained solution was transferred to a 50 mL Teflon-lined autoclave with an FTO immersed in the solution as the growth substrate. The reaction was maintained at 120 °C for 10 h.

The CoOOH was electrodeposited on FTO according to a report method[20]. The FTO with an efficient exposure area (0.283 cm²) was used as a substrate. A constant current of 0.283 mA for 100 s and subsequent 2.83 mA for 60 s was applied on FTO in a 1.0 M $CoSO_4$ aqueous solution. The as-deposited film was then activated at 1.4 $V_{RHE}$ for 100 s in 1.0 M NaOH to trigger the phase transformation from $Co(OH)_2$ to CoOOH. The final catalyst film was cleaned with deionized water for further experiments.

**Electrochemical measurements**
All electrochemical experiments were performed on a Bio-Logic SP-200 potentiostat in a standard three-electrode system using FTO glass (contact area as 0.283 cm²) as a working electrode, platinum gauze as a counter electrode, and Hg/HgO as reference electrode, respectively.

All potentials reported in this work have been converted to the reversible hydrogen electrode (RHE). All electrochemical experiments were carried out at room temperature (-298 K) in 1.0 M NaOH (pH 13.9, measured by an Ohaus Starter 2100 pH meter with temperature calibration). The cyclic voltammetry (CV) was recorded from 0.5 to 1.75 $V_{RHE}$ at a scan rate of 10 mV s$^{-1}$ unless specified. The overpotentials ($\eta$) were obtained using $\eta = E$ (vs RHE) − 1.23 V at 10 mA cm$^{-2}$ unless specially indicated. Electrochemical impedance spectroscopy (EIS) was

measured with a frequency scan range from 100 kHz to 100 MHz, and the amplitude of the sinusoidal wave was 10 mV. The overpotential ($\eta$) of CoOOH and Co-PH were obtained after $iR$ correction. The electrochemically active surface area (ECSA) of Co-PH was estimated from the electrochemical double-layer capacitance ($C_{dl}$) according to ECSA = $C_{dl}/C_s$. The $C_{dl}$ was determined by measuring a series of CVs in the non-Faradic region at a scan rate of 0.1, 0.2, 0.3, 0.4, and 0.5 V s$^{-1}$. The $C_s$ of 0.04 mF cm$^{-2}$ was taken to estimate ECSA according to a previous report[72]. The mass activity ($I_m$) was calculated according to the equation: $I_m = I / m_{Co}$, where the $m_{Co}$ was obtained by ICP-MS. Turnover frequency (TOF) was calculated according to the equation: TOF = ($j \times$ A) / ($4 \times$ F $\times n_{Co}$), where the $n_{Co}$ is the Co atom number derived from the ICP-MS results.

## In situ UV-Vis

In situ UV-Vis spectra were recorded on a QE Pro UV-Visible spectrometer (Ocean Optics) equipped with an HL-2000 light source (360 to 1100 nm). The light source was connected to the spectrometer via a fiber-optic cable (200 μm fiber core diameter). The FTO with a fixed area (0.283 cm$^2$) was used as the working electrode, and Ag/AgCl and Pt electrodes were used as the reference electrode and the counter electrode, respectively. The UV-Vis spectra of the deposited catalysts were subtracted by the background spectrum of cleaned FTO immersed in the same electrolyte. The potential-dependent UV-Vis spectra were subtracted by the absorption spectra of catalysts film at 0.5 V$_{RHE}$. According to the previous reports[51,52], the differential absorbance (dAbs./dV) of pc-CoOOH and pf-CoOOH were obtained at 730 nm. The dAbs. represents the differences in the absorbance at two consecutive potentials separated by 0.05 V, and the dV is 0.05 V.

## In situ electron paramagnetic resonance (EPR)

In situ EPR measurements were conducted at 9.73 GHz, 2 mW, and room temperature (298 K) on an X-band EPR (Bruker EMXmicro-6/1) equipped with a customized three-electrode system: Au (working electrode), Pt (counter electrode), and silver wire as a pseudo reference electrode (the potential was converted to RHE). 1 mL Co$^{II}$(phen)$_2$(OH)$_2$-containing 1.0 M NaOH was used as the electrolyte and a constant potential of 1.7 V$_{RHE}$ was applied to the working electrode. The in situ EPR data were acquired without subtraction of the background.

## Electrochemical quartz crystal microbalance (EQCM)

In situ EQCM experiments were implemented on an SRS QCM 200 instrument with an Au/Ti crystal (SRS QCM Crystal, 0100Rx3, p/n 6-00615, 5 MHz) as the working electrode. The deposition was conducted at 1.7 V$_{RHE}$ in Co$^{II}$(phen)$_2$(OH)$_2$-containing 1.0 M NaOH at room temperature. The deposited induced frequency change (Δf) was converted into the corresponding mass difference (Δm) according to the following formula[73,74].

$$\Delta f = - C_f {}^* \Delta m \qquad (1)$$

where Δf is the frequency (Hz) change, Δm is the deposited catalyst mass per unit area (g cm$^{-2}$), $C_f$ is the sensitivity factor of the crystal (56.6 Hz μg$^{-1}$ cm$^2$ for 5 MHz Au/Ti quartz crystal at room temperature).

## In situ X-ray absorption fine structure (XAFS) measurements

All the XAFS data were collected at the BL11B beamline of the Shanghai Synchrotron Radiation Facility (SSRF). The beam current of the storage ring was 220 mA in a top-up mode and the incident photons were monochromatized by a Si (111) double-crystal monochromator, with an energy resolution ΔE/E ~2×10$^{-4}$. The spot size at the sample was ~200 μm × 250 μm (H × V). The position of the absorption edge ($E_0$) was calibrated by using Co foil. The images of the in situ XAFS equipment and the electrochemical cell were shown in Supplementary

Fig. 8e, f. To preclude the influence of the FTO and obtain the in situ XAFS data, carbon paper was used as a substrate. All XAFS spectra were collected in fluorescence mode. To track the changes in the catalysts during the OER process, different anodic potentials were applied to the catalysts in 1.0 M NaOH in a typical three-electrode system. After the corresponding current density reaches a steady state, the XAFS data can be collected at the indicated potentials. All XAFS data were analyzed by the ATHENA and ARTEMIS modules implemented in the IFEFFIT software package[75].

## Physical and chemical characterizations

All samples were cleaned with deionized water and dried with nitrogen before the characterization. The ex situ UV-Vis spectra were recorded on a QE Pro UV-Visible spectrometer (Ocean Optics) equipped with a DH-2000-BAL light source (200 to 950 nm). The ex situ EPR data were acquired on an X-band Bruker EMXmicro-6/1 at 9.38 GHz, 100 K, 2 mW power. ICP-MS was performed on an Agilent 7800 spectrometer. Fourier transform ion cyclotron resonance mass spectra (FT-ICR-MS) were measured on a Bruker Solarix spectrometer equipped with a dual electrospray ionization (ESI) source in the positive ion mode. The chemical state and the Co/N ratio of the Co-PH catalyst film were analyzed by a Thermo Scientific K-alpha X-ray photoelectron spectroscopy (XPS). XPS Fitting was carried out by the Avantage software, and the binding energies were calibrated using the adventitious carbon by shifting the C 1$s$ peak to 284.8 eV. The catalyst morphology was examined by a ZEISS MERLIN Compact scanning electron microscope (SEM) with the electron gun operated at 10.0 kV. A Bruker M4 Tornado X-ray fluorescence (XRF) with a Ru target was used to determine the Co content based on Co-Kα emission. Attenuated Total Reflection Fourier Transform Infrared Spectroscopy (ATR-FTIR) analyses were performed on a Perkin−Elmer Spectrum GX spectrometer in absorption mode over a scanning range of 400−4000 cm$^{-1}$. FTIR spectrum of phenanthroline power was measured on a PE Spectrum Two spectrometer in absorption mode. The morphology and structure were further analyzed on an FEI-Tecnai G2 high-resolution transmission electron microscopy (HRTEM). Besides, element mapping and selected area electron diffraction (SAED) was conducted on an FEI TECNAL G2 F30 field emission transmission electron microscope. The sample for taking the cross-section STEM images of Co-PH deposited on FTO was prepared by ion-milling and polishing (PIPS II, GATAN) at grazing incidence mode (<5°). The corresponding STEM and element mapping images were obtained on a JEOL JEM2100F transmission electron microscope equipped with an EDS detector (Oxford Instruments). The real-time Faradaic efficiency of evolved O$_2$ during the in situ deposition of Co-PH at 10 mA cm$^{-2}$ is determined by monitoring the oxygen concentration in the headspace of gas-tight H-cell electrolyzer with an online Agilent 8890 GC gas chromatography (equipped with a GS-GASPP0 column).

## Computational details

All DFT calculations were carried out within the projected augmented wave method, as implemented in the Vienna Ab-initio Simulation Package. To generate highly accurate electrochemical stability diagrams, we employ a recently developed approach[76], which includes the use of a Hubbard U term, a van der Waals functional (optPBE)[77], and the use of spin polarization for the calculations. The $U$ value applied to d-orbitals of Co is taken as 3.50 eV. For cell shape and volume relaxations of (hydroxy)oxide compounds, a cutoff energy of 500 eV is used for the plane wave expansion. Monkhorst−Pack k-point grids are used for Brillouin zone integration. A (3 × 3 × 4) k-point grid is employed for the phen insertion within layered Co-based (oxy)hydroxides compounds with 1 ML phen, and (3 × 3 × 2) k-point grid for 1/3 ML and 2/3 ML phen, respectively. For the other bulk and surface calculations, equivalent or denser k-point grids are utilized. An orthorhombic box (30 × 30 × 30) Å$^3$ and a single k-point (1 × 1 × 1) for the Brillouin zone sampling are used for Co(phen)$_2$O$_x$H$_y$ species. The equilibrium

geometries are obtained when the maximum atomic forces are smaller than 0.02 eV/Å and when a total energy convergence of $10^{-5}$ eV is achieved for the electronic self-consistent field loop. The thermodynamic correction and solvation energy used in the free energy diagrams calculations have shown in Supplementary Table 2. We consider the following four electron reaction paths of OER with the potential $U_{RHE}$ as follows:

$$OH^* + OH^- \rightarrow O^* + H_2O + e^- \qquad (2)$$

$$O^* + OH^- \rightarrow OOH^* + e^- \qquad (3)$$

$$OOH^* + OH^- \rightarrow O_2 + V^* + H_2O + e^- \qquad (4)$$

$$V^* + OH^- \rightarrow OH^* + e^- \qquad (5)$$

Where the symbol * indicates the active sites. The corresponding reaction-free energies are calculated as follows:

$$\Delta G_1 = \Delta G_{O^*} - \Delta G_{OH^*} - eU \qquad (6)$$

$$\Delta G_2 = \Delta G_{OOH^*} - \Delta G_{O^*} - eU \qquad (7)$$

$$\Delta G_3 = \Delta G_{O2} - \Delta G_{OOH^*} - eU \qquad (8)$$

$$\Delta G_4 = \Delta G_{OH^*} - eU \qquad (9)$$

The sum of $\Delta G_1$ to $\Delta G_4$ is fixed at 4.92 eV, and $\Delta G_{OH^*}$, $\Delta G_{O^*}$, and $\Delta G_{OOH^*}$ are calculated through the following reactions.

$$H_2O + ^* \rightarrow OH^* + 0.5H_2 \qquad (10)$$

$$H_2O + ^* \rightarrow O^* + H_2 \qquad (11)$$

$$2H_2O + ^* \rightarrow OOH^* + 1.5H_2 \qquad (12)$$

The theoretical overpotential could be obtained by evaluating the difference between the maximum gap of the four intermediate states and the ideal 1.23 eV.

The charge density differences of Co-PH are considered according to the previous reports[78,79], and calculated as follows: $\Delta\rho = \rho(pc\text{-}CoO_2) - \rho(CoO_2) - \rho(phen)$.

## Data availability
All relevant data are provided in this article and its Supplementary Information.

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

## Acknowledgements

C.C. acknowledges the funding support from the Natural Science Foundation of China (22072013). J.L. acknowledges the financial support from China Scholarships Council (No. 202008455017). C.L. acknowledges the funding support from the Natural Science Foundation of China (22202034). A.M. and M.V. acknowledge the financial support from the Slovenian Research Agency (research core funding No. P2-0412 and project No. J2-2498). The XAFS beam time was granted by the BL11B end station of Shanghai Synchrotron Radiation Facility, Chinese Academy of Sciences. The staff members of BL11B are acknowledged for their support in measurements and data analyses.

## Author contributions

C.C. conceived this study and led the project. Q.W. carried out the experiments and M.X. repeated the experiments. J.L.(Junwu Liang), Z.Z., and J.Z. designed the computation and analyzed the results. L.L., X.Z., and W.W. implemented part of the measurements. C.L., M.X., Z.L., J.L.(Jiong Li), and Q.W. contributed to the XAFS measurements. C.L. and Q.W. implemented the EXAFS simulation. A.M. and M.V. performed the STEM measurements. H.-W.L., H.L., Q.W., and C.C. wrote the manuscript. All authors commented on the manuscript.

## Competing interests

The authors declare no competing interests.
