## [Peer review file · Nature Communications]

REVIEWER COMMENTS

Reviewer #1 (Remarks to the Author):

Non-covalent ligand-oxide interaction promotes oxygen evolution

Addressing only the computational part of this manuscript, the approach of the authors is sound and corresponds to the state-of-the-art. I do not have any criticism on the reported results. Yet, for publication in Nature Communications, I ask the authors to model the OER cycle to connect the adsorption free energies of the intermediate states to catalytic activity. This could contribute to gain a better understanding of the high activity of the reported molecular catalyst.

Reviewer #2 (Remarks to the Author):

Wu et al investigated Co-oxyhydroxides operated as a catalytic electrode material in alkaline OER. They investigate the influence of phenanthroline (phen) and report increased stability and low overpotentials with phen being present in electrolyte and catalyst film. The phen influence likely has not been investigated before and thus represents a novel approach.

This reviewer is sympathizing with the broad experimental approach and sees some merits of the study. However, taking into account the below list of questions and deficits regarding both significance of the author's conclusions and central technical aspects, publication cannot be recommended.

In parts this investigation provides valuable and convincing experimental evidence, specifically regarding the increased stability of the catalyst material when operated in a phen-containing electrolyte. I understood that this stability enhancement is explained as resulting from non-covalently bound phen which enhances the propensity for Co^{4+} formation and thereby the efficiency for self-healing by redeposition of Co^{2+} ions. Arguing that phen is non-covalently bound to (or within) the catalyst materials likely has led to the title suggesting on 'non-covalent ligand-oxide interactions' that promote OER.

It is a weak point in the author's argumentation that neither the non-covalent binding mode nor the intactness of catalyst-internal phen is convincingly shown. Therefore and because of lacking investigation of further compounds aside from phen, the title (and thus the main message of the study)

is not sufficiently well backed up by the findings of the study. (Moreover, 'non-covalent ligand-oxide interaction' is an unclear expression that does not describe the suggested situation well.)

A second especially critical point is the suggested performance increase by phen. An overpotential of 216 mV per cm² is reported (at 10 mA/cm²). However, this low overpotential is present only after about 2 weeks of catalyst operation. Within the first hours of catalyst operation, however, the overpotential is much higher, exceeding 500 mV, as visible in Fig. S44. In the context of a study on the phen influences, publication would require convincing elucidation of the reasons of the dramatic performance increase that requires >10 days for completion. The role of uncontrolled Fe contaminations needs to be examined and/or thoroughly excluded. In addition, it will be important to clarify for all reported data the respective status of the catalyst material regarding its long-term activation state.

Further problematic points:

- Co-O distances of 1.5 Å and 2.5 Å are reported, based on the EXAFS analysis. The first value is unreasonably short (might be in order for Co=O, but massive formation of terminal oxo species is unlikely). And the second value is unreasonably long. Similarly, also the coordination numbers are not especially reasonable. Moreover, insufficient details on the EXAFS simulation are provided. Also for submission to another journal, the EXAFS analysis would need to be approached anew. Earlier studies by others may provide guidelines, e.g. Risch et al, Energy Env. Science, 2015. Also, it should not be overlooked that the EXAFS cannot discriminate between oxygen and nitrogen ligation.

- In Fig. 2, a shift or redox waves is concluded from the CV data in Fig. 3. The underlying (mathematical) rationale needs to be explained. Right now it appears as if more or less arbitrary numbers are written into the CV figure. The shift in redox potentials is not visible in panels e and f, or is it? Minor points: What exactly is shown in Figure 3d and what does it mean?

- In Figure 3 and 4, there are panels showing lines and the data is called a phase diagram. Some unexplained ΔG value is plotted versus the applied potential. Figure caption does not say what ΔG is plotted, but just say that a phase diagram is shown. In any event, these are NOT phase diagrams (where lines typically separate areas of different phases).

- Densities of states are shown in Fig. 3c, with two red lines and two grey lines. What exactly is shown and what do these curves mean (why are they shown)?

- It is not clear whether the shown X-ray data for the 1.7 V vs. RHE catalysts has been obtained by operando spectroscopy, that is, whether the data was collected during application of the electrochemical potential. This should be stated very clearly. If it were not an operando experiment, this would weaken the impact of the data significantly.

- In the Supporting information, all figure captions need to contain clearly more information about the shown data.

- In the structures shown in Fig. S18, the Co ion coordinated to a single phen are assumed to be four-coordinate. Why is six-coordinate Co not considered?

- In Fig. S21, why should 16O-18O exchange optical spectra. Maybe the observed difference occurred just by chance (irreproducibility of, e.g., background absorption).

- In Fig. S24: It is doubtful that the assumed non-covalent coordination mode would result in disappearance of the sharp phen lines of the phen IR spectrum. The formation of broad bands in Co-PH could be reconciled more easily with a structure where phen or phen fragments coordinated to Co ions.

Reviewer #3 (Remarks to the Author):

In this work, the authors alter the ligand environment of Co to enhance the OER activity. The scientific results are interesting. The authors use a range of characterisation techniques as well as theoretical calculations to support their claims. However, there are still some issues, which I have highlighted below:

- The authors suggest an increase in coordination number from 6.4 at OCP to 7.9 at 1.7 V for pc-CoOOH. From my understanding, Co is octahedrally coordinated – what does a coordination number of ~8 mean in this case?
- In Figure 2a, it seems like the pf-CoOOH does not get oxidized at all at 1.7 V compared to OCV – this seems inconsistent with the literature and the authors other data sets. Can the authors please explain this?
- The DFT results suggest that the Co oxidation potential should decrease with an increase in phen content. However, based on the cyclic voltammograms and UV vis data in Figure 2c, d, e, it seems like only the density of Co⁴⁺ states increases, while there is no shift in the oxidation potential. Can the authors please comment on this difference?
- What is the relation between Co⁴⁺ site density and OER activity? A simple increase in the number of sites participating in the reaction cannot account for the big improvement in activity. Do the authors imagine a higher order interaction to play, like that reported recently for Ni based oxides?
<https://www.nature.com/articles/s41467-019-13061-0>,
<https://pubs.acs.org/doi/full/10.1021/jacs.1c08152>
- I think the article would benefit from some insight into why this particular ligand was chosen? Why does the coordination of the Co site with this specific ligand result in it being more oxidized? It is unclear to me how and why the authors choose this ligand? What is the impact of this ligand on the binding energetics of oxygenated intermediates which are key to describe OER activity?
- Does the electrolyte for the Co-PH data in Figure 6 contain phen? In other words, is the Co-PH catalyst stable in electrolyte that does not contain the phen ligand?
- It is unclear if the Co sites that interact with the phen ligands are still able to bind reaction intermediates of OER or not?

Dear Reviewers:

We highly appreciate your careful review and constructive comments which help us to significantly improve the quality of this manuscript (NCOMMS-22-17715). Now we have carefully considered all your comments and revised the manuscript accordingly. We made corresponding corrections point by point as follows (corrections in the revised manuscript are marked in blue color).

Reviewer #1 (Remarks to the Author):

Non-covalent ligand-oxide interaction promotes oxygen evolution. Addressing only the computational part of this manuscript, the approach of the authors is sound and corresponds to the state-of-the-art. I do not have any criticism on the reported results. Yet, for publication in Nature Communications, I ask the authors to model the OER cycle to connect the adsorption free energies of the intermediate states to catalytic activity. This could contribute to gain a better understanding of the high activity of the reported molecular catalyst.

Responses 1:

- Thanks very much for the positive evaluation of our work and your constructive comments. It is a good idea to link the catalytic activity with the adsorbed intermediates to model the OER cycle.
- As suggested, we supplemented the reaction free energies of the intermediates on Co-PH and CoO₂ via DFT calculations (Fig. R1). The non-covalent interaction between phen and CoO₂ reduces the reaction free energy from OH* to O* (rate-limiting step) from 1.98 to 1.63 eV, while the following steps remain similar.
- In this revised manuscript, we also added the charge density differences of Co-PH and reaction free energies of the other eight Co sites in the *main text* and *supplementary information*. These results suggest that the non-covalent interaction is not localized on a specific Co site.
- The supplementary DFT calculations are shown in Fig. 7 on page 14 in the *main text*, and in Supplementary Figs. 52-65 (not shown here) on pages S60-S73 in the *supplementary information*. The corresponding discussion has been added on pages 14-15 in the *main text* as follows:

“Understanding improved activity through modeling. To better understand the role of phen, we further studied OER activity through DFT calculations. We first evaluated the steady-state configurations of CoO₂ (10-10) and Co-PH (10-10) surfaces (Fig. 7a). The calculated surface free energy diagrams of CoO₂ and Co-PH indicate that, under OER conditions, coordinatively unsaturated surface O sites are saturated with H_{ad} by forming bridge OH species, and coordinatively unsaturated Co sites are saturated with OH_{ad} by forming atop OH* (Fig. 7b and Supplementary Fig. 52). Thus, OER starts from the deprotonation of the surface OH*, as has been proposed in the previous reports^{16,23,70}. Also, we found that OH* deprotonation to O* is the potential limiting step for both bridge OH* and atop OH* (Fig. 7d and Supplementary Fig. 53). For the deprotonation of bridge OH*, the overpotential is about 0.2~0.6 V lower than that of atop OH* (Supplementary Fig. 54). Thus, OH bridged dual Co-Co sites are reaction centers.*

For CoO₂, the overpotential is 0.75 V (Fig. 7d). The presence of polaron-like Co³⁺ induced by phen, however, can reduce the overpotential to 0.4~0.5 V, which is dependent on the relative distance of the reaction center to the location of Co³⁺ site (Fig. 7c-7d and Supplementary Figs. 55~65). The reaction center with the lowest overpotential of 0.4 V is the one with Co³⁺ forming the Co³⁺-Co⁴⁺ reaction center. This type of reaction center facilitates the potential-limiting deprotonation of OH to O* which is accompanied by the oxidation of Co³⁺ to Co⁴⁺, as observed by the change of Co magnetic moment from 0 to 1 μB (Supplementary Figs. 58-61). However, even if the Co³⁺ site is not part of the reaction center, i.e., forming Co⁴⁺-Co⁴⁺ type reaction*

centers, the overpotential only increases slightly (Supplementary Figs. 62-65). For example, for the cases of Co^{3+} with one atom away from the reaction center, the overpotential only increases 0.02-0.05 V due to the effect of polaron, i.e., Co^{3+} is still oxidized to Co^{4+} during the deprotonation of OH^* to O^* (Supplementary Figs. 58-61). The feature of polaron is more clearly illustrated in the cases where Co^{3+} is far away from the reaction center, e.g., on the opposite side of the slab model. For these cases, the overpotential is only up to 0.1 V higher (Supplementary Figs. 62-65). These results highlight the importance of non-covalent ligand-oxide interactions in reducing the OER overpotential by generating abundant Co^{4+} , active polarons, and dynamic reaction centers.”

Fig. R1. OER mechanism. (a) Surface free-energy diagram of CoO₂ and Co-PH with respect to the surface saturated with hydrogen adsorption (H_{ad}) on bridge O (O_{br}) and OH adsorption (OH_{ad}) on coordinatively unsaturated sites. (b) The structures of different surface phases on Co-PH, and adsorbates of surface phases are highlighted by light blue circles on the top views. (c) The top view (upper) and side view (under) of charge density differences of Co-PH that surface saturated with H_{ad} and OH_{ad}, that is, H_{ad}+OH_{ad} (OH^{*}). The purple isosurfaces represent an increase in the total charge density, and the orange isosurfaces represent a decrease in the total charge density. (d) The reaction free-energy diagrams for OER on CoO₂ and Co-PH at 0 V_{RHE}. Insets are the corresponding enlarged models.

Reviewer #2 (Remarks to the Author):

Wu et al investigated Co-oxyhydroxides operated as a catalytic electrode material in alkaline OER. They investigate the influence of phenanthroline (phen) and report increased stability and low overpotentials with phen being present in electrolyte and catalyst film. The phen influence likely has not been investigated before and thus represents a novel approach.

This reviewer is sympathizing with the broad experimental approach and sees some merits of the study. However, taking into account the below list of questions and deficits regarding both significance of the author's conclusions and central technical aspects, publication cannot be recommended.

We highly appreciate the positive comment and previous professional suggestions from the reviewer. The comments are quite helpful, please find our response and revision as follows.

In parts this investigation provides valuable and convincing experimental evidence, specifically regarding the increased stability of the catalyst material when operated in a phen-containing electrolyte. I understood that this stability enhancement is explained as resulting from non-covalently bound phen which enhances the propensity for Co^{4+} formation and thereby the efficiency for self-healing by redeposition of Co^{2+} ions. Arguing that phen is non-covalently bound to (or within) the catalyst materials likely has led to the title suggesting on 'non-covalent ligand-oxide interactions' that promote OER.

It is a weak point in the author's argumentation that neither the non-covalent binding mode nor the intactness of catalyst-internal phen is convincingly shown. Therefore and because of lacking investigation of further compounds aside from phen, the title (and thus the main message of the study) is not sufficiently well backed up by the findings of the study. (Moreover, 'non-covalent ligand-oxide interaction' is an unclear expression that does not describe the suggested situation well.)

Responses 1:

- We are thankful to the reviewer for pointing out this concern. We are sorry that we did not clearly describe the key points leading to misleading.
- Regarding the non-covalent interactions between phen with Co oxide, we made the conclusion based on experimental evidence and DFT calculations.

(1) We first demonstrated valence-dependent coordination. As shown in Fig. R2, the Co^{2+} sites in $\text{Co}(\text{OH})_2$ can coordinate with phen and dissolve into the phen-containing 1.0 M NaOH at OCP (Fig. R2a), while the Co^{3+} sites in CoOOH are stable on the electrode and cannot interact with phen (Fig. R2b). To further confirm this valence-dependent interaction, applying a potential on CoOOH at $0.7 V_{\text{RHE}}$ to reduce to $\text{Co}(\text{OH})_2$ in phen-containing 1.0 M NaOH also causes the coordination-induced dissolution of CoOOH . These results suggest that the phen can coordinate with Co^{2+} rather than Co^{3+} .

(2) Upon the anodic polarization at $1.7 V_{\text{RHE}}$ (partial Co^{3+} sites can be oxidized to Co^{4+}), the phen can be involved in the catalyst layer (Fig. R3a-3b), and the pc- CoOOH catalyst presents higher OER activity relative to pf- CoOOH (Fig. R3c). Thus, we reasoned that an uncovered interaction between phen and CoO_2 should be present. The relative discussions are shown in Fig. 1a on page 5 and Fig. 4a on page 10 in the *main text*, Supplementary Fig. 1 on page S2, and Supplementary discussion to Supplementary Figs. 18-21 on pages S23-S24 in the *supplementary information*. We added one sentence on page S24 in the *supplementary information* as follows:

"The interplay between phen with CoO_2 is significantly different from that of phen and $\text{Co}(\text{OH})_2$ or CoOOH ".

Fig. R2. The coordination ability of Co^{2+} and Co^{3+} with phen. (a) The CV curves of the $\text{Co}(\text{OH})_2$ catalysts at OCP in phen-free (pf-) and phen-containing (pc-) 1.0 M NaOH electrolytes. (b) The CV curves of the CoOOH catalysts at OCP in phen-free (pf-) and phen-containing (pc-) 1.0 M NaOH electrolytes. (c) The CV curves of CoOOH after 2.0 h of reduction at 0.7 V_{RHE} in phen-containing NaOH, the inset are the corresponding images of electrodes. The phen concentration is 7 mM in all cases.

Fig. R3. The coordination ability of Co^{4+} with phen. (a) The element mapping of pc- CoOOH . (b) The N 1s XPS spectra of pf- CoOOH and pc- CoOOH . (c) The j - V curves of pf- CoOOH and pc- CoOOH . “pf” and “pc” represent that the pristine CoOOH pre-catalysts were treated for 10 h in phen-containing and phen-free 1.0 M NaOH electrolytes, respectively.

(3) DFT calculations were used to reveal the interactions between phen with CoO_2 . We know that the proposed “non-covalent” is an energy-based notion, thus we calculated the formation energy of CoOOH and CoO_2 containing different density phen (i.e., 1 ML, 0.5 ML, 0.33 ML phen), as shown in Fig. R4. In general, we distinguish the covalent and noncovalent interactions by the interaction distance and stabilization energy (*Chem. Rev.*, 2000, 100, 143-167). Typically, covalent interactions are short-range and covalent bonds are generally $< 2 \text{ \AA}$, yet the non-covalent interactions are $> 2 \text{ \AA}$. In our work, the shortest distance between the N atom and Co atom is $\sim 4.12 \text{ \AA}$ in H-pc- CoO_2 (Fig. R5), which should belong to the range of non-covalent interactions. Importantly, the calculated energies within the range of $64\sim 113 \text{ kJ mol}^{-1}$, depending on low(L)-, middle(M)-, and high(H)-density phen, are considerably lower than the stabilization energy of covalent bond about $200\sim 300 \text{ kJ mol}^{-1}$. The interactions with low stabilization energies and relatively long bond distances represent “non-covalent interactions” (for example the cation-based non-covalent interaction in *Nat. Chem.*, 2009, 1, 466-472). We added Fig. R5 to the *supplementary information* and more discussions on page S24 in the *supplementary information* as follows:

“The covalent and non-covalent interactions can be distinguished by the stabilization energy and distance in equilibrium. The interactions with low stabilization energies and relatively long distances are ascribed to the “non-covalent interactions”⁷. Typically, covalent interactions are short-range and covalent bonds are

generally shorter than 2 Å, and non-covalent interactions are known to act at distances > 2 Å⁸. In this work, the shortest distance between the N atom and Co atom is 4.12 Å in H-pc-CoO₂ (Supplementary Fig. 18). Moreover, the calculated interaction energies within the range of 64~113 kJ mol_(phen)⁻¹, depending on low(L)-, middle(M)-, and high(H)-density phen, which are considerably smaller than the stabilization energy of covalent bond of about 200~300 kJ mol⁻¹. Thus, the interaction between phen with CoO₂ has been considered non-covalent. This model of H-density phen is close to the phen content in the pc-CoOOH catalysts.”

Fig. R4. The formation energy of layered CoOOH and CoO₂ containing high(H)-, middle(M)-, and low(L)-density of phen (i.e., 1 ML, 0.5 ML, and 0.33 ML phen) in the interlayers.

Fig. R5. The geometric structures of layered CoO₂ containing high(H)-, middle(M)-, and low(L)-density of phen (i.e., 1 ML, 0.5 ML, and 0.33 ML phen) in the interlayers. The dark blue notes indicate the shortest distance between the N atom and the Co atom.

(4) We applied ΔG -potential diagrams to further support the conclusion. The Co(OH)₂, Co^{II}(phen)₂(OH)₂, Co^{III}(phen)₂OOH, Co^{IV}(phen)₂O₂, phen inserted CoOOH (Co^{III}-PH), and phen-inserted CoO₂ (Co^{IV}-PH) were calculated. From the diagrams in Fig. R6, upon oxidation of the Co(phen)₂(OH)₂ precatalyst, it will convert into Co^{IV}-PH, with phen non-bonding interaction at $\sim 1.37 V_{RHE}$, rather than the energy-unfavorable covalent Co(phen)₂O₂ or Co(phen)O₂ structures. Thus, electrochemical oxidation of soluble Co(phen)₂(OH)₂ leads to the formation and deposition of pc-CoO₂ fragments instead of oxidized complexes with covalent bonds.

Fig. R6. The free energy diagram of $\text{Co}(\text{OH})_2$, $\text{Co}^{\text{II}}(\text{phen})_2(\text{OH})_2$, $\text{Co}^{\text{III}}(\text{phen})_2\text{OOH}$, $\text{Co}^{\text{IV}}(\text{phen})_2\text{O}_2$, phen inserted CoOOH ($\text{Co}^{\text{III}}\text{-PH}$), and phen-inserted CoO_2 ($\text{Co}^{\text{IV}}\text{-PH}$).

- Regarding the intactness of phen in Co-PH film, as shown in the *main text* and *supplementary information*, we provided as massive evidence as we can think of to verify the existence of intact phen in the catalyst films through XPS element analysis, STEM element mapping, EDX element analysis, UV-Vis, and ATR-FTIR (Page 12 in the *main text*). Even so, we supplemented the FT-ICR-MS measurements of Co-PH films to further confirm the intactness (Fig. R7): specifically, the Co-PH film was dissolved in HNO_3 , and the obtained solution was thereafter adjusted the pH to 13.9 by adding NaOH. We found that the dissolved Co-PH exhibits dominated peaks at m/z 453.07562 and 383.12805, which are ascribed to $[\text{Co}(\text{phen})_2(\text{OH})_2]^{2+}$ and $[2(\text{phen}) + \text{Na}]^+$, respectively. Thus, we can conclude that the phen molecules still maintain intactness in the Co-PH film. The supplementary FT-ICR-MS is shown in Supplementary Fig. 47 in the *supplementary information* and the corresponding discussion has been provided on page 13 in the *main text* as follows:

“Importantly, the phen was observed intact in Co-PH films after OER operation (Supplementary Fig. 47).”

Fig. R7. The FT-ICR-MS spectra of dissolved Co-PH film. The Co-PH film was deposited at 1.7 V_{RHE} in $\text{Co}(\text{phen})_2(\text{OH})_2$ -containing 1.0 M NaOH for 10 h. For the measurements of FT-ICR-MS, typically, deposited Co-PH film was first soaked in the 10 wt% HNO_3 solutions over 10 h to dissolve. Excess NaOH solid was then added to this solution to adjust the solution pH, resulting in the formation of 1.0 M NaOH (pH 13.9). The obtained solution was used for the FT-ICR-MS characterization. The spectra were recorded using dual electrospray ionization (ESI) source in the positive ion mode.

- In this work, we selected phen as a model ligand to explore the nature of the ligand-promoted OER activity. Before the selection of phen, we screened a series of ligands (from 1 N to 4 N). As shown in Fig. R8, all ligands show enhanced OER activity in different degrees, but phen performs best. As demonstrated in this work, proving the non-covalent interaction requires a lot of evidence, especially time-consuming DFT calculations, so we leave other ligands for further study. We believe that this work could provide references for other ligand-based systems and offer a dynamic view for understanding the ligand-oxide interaction.

Fig. R8. The j - V curves of CoOOH and ligand-treated CoOOH. The ligand-treated CoOOH represents that the pristine CoOOH pre-catalysts were treated at 1.7 V_{RHE} for 10 h in ligand-containing 1.0 M NaOH electrolytes. The structure inset in the figures indicates the corresponding ligand. The grey j - V curves in all figures indicate the pristine CoOOH.

- We have carefully re-considered the title and the main message we have delivered. The goal is to clarify the key role of non-bonding ligand on oxides/hydroxide for the OER activity. This kind of non-covalent interaction has been ignored for a long time, especially in the molecular system, where the N-Metal bond may break under OER conditioning, but the ligand may still contribute to the final OER activity. This work should be urgent to fill that knowledge gap.

A second especially critical point is the suggested performance increase by phen. An overpotential of 216 mV per cm^2 is reported (at 10 mA/cm^2). However, this low overpotential is present only after about 2 weeks of catalyst operation. Within the first hours of catalyst operation, however, the overpotential is much higher, exceeding 500 mV, as visible in Fig. S44. In the context of a study on the phen influences, publication would require convincing elucidation of the reasons of the dramatic performance increase that requires >10 days for completion. The role of uncontrolled Fe contaminations needs to be examined and/or thoroughly excluded. In

addition, it will be important to clarify for all reported data the respective status of the catalyst material regarding its long-term activation state.

Responses 2:

- Thanks very much for this question. We present both the catalyst deposition process and the subsequent OER stability test in one curve to better understand the whole process and activity variation trend. Within the first few hours, the catalyst deposition self-optimizes to the best state. Based on the Co-PH mass-OER activity relationship, the enhanced OER activity is highly relevant to the mass loading. Finally, its OER activity tends to saturate. This is different from the pre-loaded catalyst.
- The catalyst deposition rate also depends on the current densities or the applied potentials. A fast catalyst deposition can be obtained at a higher applied potential. At the beginning of catalyst electrodeposition (Supplementary Fig. 49, and Fig. 6b), the overpotential is high owing to much less mass load. The Co-PH was *in-situ* deposited on bare FTO and operated in $\text{Co}(\text{phen})_2(\text{OH})_2$ -containing 1.0 M NaOH. The initial overpotential hence indicates the activity of bare FTO in $\text{Co}(\text{phen})_2(\text{OH})_2$ -containing 1.0 M NaOH. As shown in Fig. R9, the bare FTO in $\text{Co}(\text{phen})_2(\text{OH})_2$ -containing 1.0 M NaOH exhibits poor activity, with an overpotential of over 600 mV at 10 mA cm^{-2} . Despite this, the Co-PH could quickly deposit on bare FTO, which decreases the overall overpotential from over 600 to 216 mV at 10 mA cm^{-2} . To clearly show the operation condition, we added one sentence on page 13 in the *main text* and in the figure caption of Fig. 6 in the *main text*, respectively, as follows:

“The first a few hours of operation can be considered as catalyst deposition, and this process was involved in stability test for understanding the activity trend.”

“The stability test of as-prepared CoOOH was operated in 1.0 M NaOH. Co-PH was *in-situ* deposited on bare FTO and its stability test was operated in $\text{Co}(\text{phen})_2(\text{OH})_2$ -containing 1.0 M NaOH.”

Fig. R9. The j -V curves of bare FTO in $\text{Co}(\text{phen})_2(\text{OH})_2$ -containing 1.0 M NaOH. The catalyst deposition should happen even at the first scan although the load should be very low.

- Thanks very much for raising this concern about the possible iron contamination. We always take it very seriously after the published works by the Boettcher group. To eliminate iron contamination, we have established reliable test protocols for electrochemical experiments. In all experiments, the high-purity metal salts were used (actually highest purity commercially available in Adamas), and especially, the high-purity NaOH was further purified to remove the impurities according to the previous report (*J. Am. Chem. Soc.*,

2015, 137, 3638-3648). In addition, we supplemented XRF measurements to test the Co-PH films after different times on FTO. As shown in Fig. R10, we didn't observe any Fe K α signal. Thus, iron contamination was not involved.

Fig. R10. The XRF data of Co-PH after different operation time periods.

Further problematic points:

- Co-O distances of 1.5 Ang and 2.5 Ang are reported, based on the EXAFS analysis. The first value is unreasonably short (might be in order for Co=O, but massive formation of terminal oxo species is unlikely). And the second value is unreasonably long. Similarly, also the coordination numbers are not especially reasonable. Moreover, insufficient details on the EXAFS simulation are provided. Also for submission to another journal, the EXAFS analysis would need to be approached a new. Earlier studies by others may provide guidelines, e.g. Risch et al, Energy Env. Science, 2015. Also, it should not be overlooked that the EXAFS cannot discriminate between oxygen and nitrogen ligation.

Responses 3:

- Thanks very much for careful review and pointing out this question. We have carefully double-checked EXAFS parameter fitting for the Co samples, the Co-O and Co-Co distance should be corrected as 1.93 Å and 2.89 Å in pf-CoOOH and pc-CoOOH, respectively, according to the latest EXAFS fitting results (Fig. R11-R12). And the coordination numbers of Co-O and Co-Co are also corrected.

Fig. R11. (a) Fitting curves of Co K-edge EXAFS in R spaces for pf-CoOOH and (b) for pc-CoOOH.

Fig. R12. Fitting curves of Co K-edge EXAFS in R spaces for pf-CoOOH at (a) OCP and (b) at 1.7 V_{RHE} . Fitting curves of Co K-edge EXAFS in R spaces for pc-CoOOH at (c) OCP and (d) at 1.7 V_{RHE} .

- All EXAFS analysis has been updated, please find the details on Supplementary Fig. 4 on page 5, Supplementary Fig. 8-9, and Supplementary Table 1 on page 9-11 in the *supplementary information*. The corresponding discussion was added to page 6 in the *main text* as follows:

“The k^3 -weighted Fourier-transformed Co K-edge extended X-ray absorption fine structure (EXAFS) reveal a Co-O distance of 1.93 Å and a Co-Co distance of 2.89 Å^{46,47}. While the Co-O and Co-Co distances of pc-CoOOH and pf-CoOOH keep the same, pc-CoOOH exhibited a relatively lower Co-Co coordination number (Supplementary Figs. 4b-4d).”

“In addition, the Co-O coordination number increased from 3.8 at OCP to 5.0 at 1.7 V_{RHE} for pc-CoOOH relative to a slightly increase for pf-CoOOH⁵⁰ (Fig. 2b, Supplementary Figs. 8-9 and Supplementary Table 1).”

- In this work, the reference (*Energy Environ. Sci.*, 2015, 8, 661-674) has been carefully studied and cited in an earlier manuscript, now ref. 46 on page 6 in the *main text*. Also, we appreciate the reviewer for reminding us that the EXAFS cannot discriminate between oxygen and nitrogen ligation.

- In Fig. 2, a shift or redox waves is concluded from the CV data in Fig. 3. The underlying (mathematical) rationale needs to be explained. Right now, it appears as if more or less arbitrary number are written into the CV figure. The shift in redox potentials is not visible in panels e and f, or is it? Minor points: What exactly is shown in Figure 3d and what does it mean?

Responses 4:

- Thanks very much for this question. As the CV data are shown in Fig. 2c, the onset potential for the oxidation of Co^{3+} to Co^{4+} in pc-CoOOH shifts to a lower potential of ~ 50 mV relative to pf-CoOOH. The lower onset oxidation potential of Co^{3+} to Co^{4+} suggests that the introduction of phen promotes the generation of Co^{4+} in pc-CoOOH. Besides the $\text{Co}^{2+}/\text{Co}^{3+}$ redox couple of pc-CoOOH cathodically shifts ~ 40 mV relative to the pf-CoOOH, again confirming the promoted oxidation of Co sites in the presence of phen. We rewrote the text on page 7 in the *main text* as follows:

“The $\text{Co}^{2+}/\text{Co}^{3+}$ oxidation peak of pc-CoOOH negatively shifts by ~ 40 mV relative to that of pf-CoOOH. Further, compared to pf-CoOOH, the onset potential for the oxidation of Co^{3+} to Co^{4+} on pc-CoOOH negatively shifts from 1.31 to 1.26 V_{RHE} .”

- As discussed above, the cathodic shift of redox potentials in Fig. 2c indicates the easier oxidation of Co sites at the same oxidation potential. We hence applied *in-situ* UV-Vis to track the Co^{4+} content as a function of applied bias at 730 nm for both pc-CoOOH and pf-CoOOH in 1.0 M NaOH from 1.2 to 1.6 V_{RHE} (Fig. 2d on page 7 in the *main text*). The absorption was observed between 1.3 V_{RHE} to 1.5 V_{RHE} , which coincides with the potential range where Co^{3+} oxidation to Co^{4+} starts to occur. Therefore, the much steeper differential absorbance between 1.3 and 1.5 V_{RHE} for pc-CoOOH illustrates the faster generation of Co^{4+} in the presence of phen, which agrees with the shift of the redox couple in Fig. 2c in the *main text*. Compared with the pf-CoOOH, the earlier increase in absorption was detected around ~ 1.0 V in pc-CoOOH, suggesting the lower oxidation potential of Co sites in pc-CoOOH. Besides the potential-dependent absorbance of pc-CoOOH presents ~ 1.8 times higher intensity for Co^{4+} at 730 nm relative to pf-CoOOH at 1.75 V_{RHE} , as shown in Fig. 2e on page 6 in the *main text*, further suggesting the favorable oxidation of Co^{3+} to Co^{4+} . We used EPR to verify the formation of Co^{4+} because of the easier charge accumulation in pc-CoOOH, therefore, the shift in redox potentials also is confirmed in Fig. 2f.
- Fig. 2d shows the difference in differential absorbance at 730 nm between 1.3 and 1.5 V_{RHE} of pf-CoOOH and pc-CoOOH. Here the differential absorbance at 730 nm represents the potential-dependent population of Co^{4+} . Thus, the much steeper differential absorbance between 1.3 and 1.5 V_{RHE} for pc-CoOOH illustrates the faster generation of Co^{4+} in the presence of phen.

Regarding the calculation methods of differential absorbance in Fig. 2d, we supplemented the details in the experimental section on page 19 in the *main text* as follows:

“According to the previous study^{55,56}, the differential absorbance ($d\text{Abs.}/dV$) of pc-CoOOH and pf-CoOOH were obtained at 730 nm. The $d\text{Abs.}$ represents the differences in the absorbance at two consecutive potentials separated by 0.05 V, and the dV is 0.05 V.”

- In Figure 3 and 4, there are panels showing lines and the data is called a phase diagram. Some unexplained delta-G value is plotted versus the applied potential. Figure caption does not say what delta-G is plotted, but just say that a phase diagram is shown. In any event, these are NOT phase diagrams (where lines typically separate areas of different phases).

Responses 5:

- Thanks very much for this comment. Here we used ΔG -potential diagram to evaluate the stability of $\text{Co}(\text{OH})_2$, CoOOH and CoO_2 in the presence/absence of phen. The potential is the applied thermodynamic potential, and ΔG represents the corresponding reaction free energy. We have corrected the words “phase diagram” to

“free energy diagram”, and hope it would be clear now.

- We appreciate the reviewer for reminding us for distinguishing the typical potential–pH phase diagrams.

- Densities of states are shown in Fig. 3c, with two red lines and two grey lines. What exactly is shown and what do these curves mean (why are they shown)?

Responses 6:

- Thanks very much for your questions. Fig. 3c shows the density of states near the Fermi level of pf-CoO₂ and the high density of phen-embedded CoO₂. We calculated the densities of states to study the charge transfer kinetics of CoO₂ in the presence/absence of phen. As discussed on pages 8-9 in the *main text*, we know the phen could facilitate the conversion of CoOOH to CoO₂. However, bulk CoO₂ is a semiconductor with high resistance, resulting in unfavorable charge transfer in the catalytic process, and thus greatly restricting its OER activity. To explain the phen-promoted catalytic activity, we calculated the densities of states of pf-CoO₂ and the high density of phen-embedded CoO₂. We found that phen embedment increases the density of states near the Fermi level, which would improve the charge transfer kinetics of high-valence Co. The improved charge transfer also is one of the reasons why phen could promote water oxidation activity.

- It is not clear whether the shown X-ray data for the 1.7 V vs. RHE catalysts has been obtained by operando spectroscopy, that is, whether the data was collected during application of the electrochemical potential. This should be stated very clearly. If it were not an operando experiment, this would weaken the impact of the data significantly.

Responses 7:

- Thanks very much for this concern. In this work, FTO has been used as the substrate throughout the measurements. Therefore, we also resort to FTO as the substrate in XAFS measurements. We first tried to obtain XAS data through the beamline setup, which is shown in Fig. R13. However, the FTO substrate inevitably absorbs the X-rays, so *in-situ* measurement cannot be enabled. Alternatively, to evaluate how phen stabilizes Co valence state monitor the changes of catalyst on FTO substrate, we applied a quasi-*in-situ* experimental approach. Typically, different anodic potentials on the catalysts in a typical three-electrode system for 10 min, and thereafter the side of FTO with catalyst was quickly for XAFS and the data was recorded immediately by fluorescence mode. We added one sentence on page 20 in the *main text* to make the experiment methods clear as follows:

“After the electrochemical treatment, the FTO with catalyst was quickly measured. Thereafter, the XAFS data were collected immediately by fluorescence modes.”

- From the XAFS discussions of Fig. 2a-2b on page 6 in the *main text*, we can conclude the conclusion that phen could promote the Co valence toward to higher oxidation state. Importantly, we carried out subsequent electrochemical tests, XPS, EPR and *in-situ* EPR tests as well as *in-situ* UV-Vis measurements. Various experiments overall demonstrate the above key points.

Fig. R13. The schematic of XAFS equipment.

- In the Supporting information, all figure captions need to contain clearly more information about the shown data.

Responses 8:

- Thanks very much for your suggestions. We have double checked all figure captions and supplemented more details. All revised texts are marked in blue color.

- In the structures shown in Fig. S18, the Co ion coordinated to a single phen are assumed to be four-coordinate. Why is six-coordinate Co not considered?

Responses 9:

- Thanks very much for this interesting question. Both the four-coordinated and six-coordinated structures have been discussed in this work. As shown in Supplementary Fig. 22 on page S25 in the *supporting information*, we constructed six phen-coordinated structures to simulate the electrodeposition process from the homogenous electrolyte to a heterogenous catalyst. Considering the oxidation process of six-coordinated $\text{Co(phen)}_2(\text{OH})_2$ molecule, higher valent Co centers, like $\text{Co(phen)}_2\text{OOH}$ and $\text{Co(phen)}_2\text{O}_2$ molecules with different Co valence, have been considered. On the other hand, the phen can dissociate from the six-coordinated structure to form possible four-coordinated Cophen(OH)_2 , CophenOOH , and CophenO_2 intermediates although energetically unfavorable.

- In Fig. S21, why should ^{16}O - ^{18}O exchange optical spectra. Maybe the observed difference occurred just by chance (irreproducibility of, e.g., background absorption).

Responses 10:

- Thanks very much for raising this question. The results are well repeatable. We have supplemented another

UV-Vis spectrum in Fig. R14. The same conclusion can be made. In addition, we supplemented the UV-Vis measurements of H_2^{18}O and H_2^{16}O in Fig. R15, which did not show a significant difference.

Fig. R14. The UV-Vis spectra of Co-phen in H_2^{18}O , D_2^{16}O , and H_2^{16}O with 1.0 M Na^{16}OH , respectively. The experiment was repeated several times and the repeated results were shown in panels (a) and panel (b). The background is deducted by the bare quartz cuvettes.

Fig. R15. (a) The UV-Vis spectra of H_2^{18}O and H_2^{16}O , respectively. (b) The difference spectrum between H_2^{18}O and H_2^{16}O . The background is deducted by the bare quartz cuvettes.

- In Fig. S24: It is doubtful that the assumed non-covalent coordination mode would result in disappearance of the sharp phen lines of the phen IR spectrum. The formation of broad bands in Co-PH could be reconciled more easily with a structure where phen or phen fragments coordinated to Co ions.

Responses 11:

- Thanks very much for this concern. Based on the discussions in Responses 1, we can know the phen could interact with charged Co oxide fragments through non-covalent coordination under OER conditioning in the *in-situ* deposited Co-PH film. Here we used the *ex-situ* ATR-FTIR spectrum to determine the presence of phen in Co-PH film. However, it doesn't accurately reflect the coordination structure between phen with Co sites owing to the *ex-situ* condition. In summary, these comments and suggestions really helped us to improve the quality of this manuscript.

Reviewer #3 (Remarks to the Author):

In this work, the authors alter the ligand environment of Co to enhance the OER activity. The scientific results are interesting. The authors use a range of characterization techniques as well as theoretical calculations to support their claims. However, there are still some issues, which I have highlighted below:

Responses 1:

- We truly appreciate your encouraging comments and insightful suggestion. We carefully considered all the comments and suggestions to further improve the quality of this manuscript as follows.
- The authors suggest an increase in coordination number from 6.4 at OCP to 7.9 at 1.7 V for pc-CoOOH. From my understanding, Co is octahedrally coordinated – what does a coordination number of ~8 mean in this case?

Responses 2:

- Thanks very much for your careful review and pointing out this question. We are very sorry for our careless mistake about the EXAFS simulation and got misleading coordination numbers. We have carefully checked EXAFS parameter fitting for all Co samples as shown in Fig. R16, and the coordination numbers are rectified on page 6 in the *main text*. We have updated the text on page 6 as follows:

“In addition, the Co-O coordination number increased from 3.8 at OCP to 5.0 at 1.7 V_{RHE} for pc-CoOOH relative to a slightly increase for pf-CoOOH⁵⁰ (Fig. 2b, Supplementary Figs. 8-9 and Supplementary Table 1).”

Fig. R16. Fitting curves of Co K-edge EXAFS in R spaces for pf-CoOOH at (a) OCP and (b) at 1.7 V_{RHE}. Fitting curves of Co K-edge EXAFS in R spaces for pc-CoOOH at (c) OCP and (d) at 1.7 V_{RHE}.

- In Figure 2a, it seems like the pf-CoOOH does not get oxidized at all at 1.7 V compared to OCV – this seems inconsistent with the literature and the authors other data sets. Can the authors please explain this?

Responses 3:

- Thanks very much for this question. We apologize for this misleading drawing of Fig. 2a on page 6 in the *main text*. As shown in Fig. R17, the Co K-edge XANES of pf-CoOOH at 1.7 V_{RHE} shifts to higher energy relative to OCP. This right shift of Co K-edge XANES toward higher energy indicates a higher Co oxidation state in pf-CoOOH at 1.7 V_{RHE}. We have modified the line thickness and style of Fig. 2a on page 6 in the *main text*, and hope that they are now clear.

Fig. R17. The enlarged Co K-edge XANES of pf-CoOOH at OCP and after 1.7 V_{RHE} polarization for 10 min.

- The DFT results suggest that the Co oxidation potential should decrease with an increase in phen content. However, based on the cyclic voltammograms and UV vis data in Figure 2c, d, e, it seems like only the density of Co⁴⁺ states increases, while there is no shift in the oxidation potential. Can the authors please comment on this difference?

Responses 4:

- Thanks very much for this comment. DFT calculation shows that the introduction of phen could decrease the oxidation potential of Co sites, while at the same potential during OER greatly elevating the population of Co⁴⁺ sites. The DFT results are well supported by our experimental results. As shown in Fig. 2c on page 6 in the *main text*, the oxidation potential of Co²⁺/Co³⁺ for pc-CoOOH cathodically shifts ~ 40 mV relative to that of pf-CoOOH. In addition, the onset potential for the oxidation of Co³⁺ to Co⁴⁺ also cathodically shifted from 1.31 to 1.26 V_{RHE}. The CV data nicely supports the DFT results. *In-situ* UV-Vis was further used to observe the redox process. As shown in Fig. 2e on page 6 in the *main text*, compared with the pf-CoOOH, the earlier increase in absorption was detected around ~ 1.0 V in pc-CoOOH, suggesting the lower oxidation potential of Co sites in pc-CoOOH. In addition, upon increasing the potential to the water oxidation region, a much steeper increase in absorption of pc-CoOOH was observed (Fig. 2d-2e on page 6 in the *main text*). In summary, the increase in Co⁴⁺ content at the same potential and the lower oxidation potential of Co sites support the DFT results. We reworded that part for CV data on page 7 in the *main text* as follows:

“The Co²⁺/Co³⁺ oxidation peak of pc-CoOOH negatively shifts by ~40 mV relative to that of pf-CoOOH. Further, compared to pf-CoOOH, the onset potential for the oxidation of Co³⁺ to Co⁴⁺ on pc-CoOOH negatively shifts from 1.31 to 1.26 V_{RHE}.”

- What is the relation between Co^{4+} site density and OER activity? A simple increase in the number of sites participating in the reaction cannot account for the big improvement in activity. Do the authors imagine a higher order interaction to play, like that reported recently for Ni based oxides?

<https://www.nature.com/articles/s41467-019-13061-0>, <https://pubs.acs.org/doi/full/10.1021/jacs.1c08152>

Responses 5:

- Thanks very much for your suggestions to inspire us to deepen our understanding. For the relation between increased Co^{4+} site density and OER activity, we summarize three points as follows: (1) truly higher Co^{4+} site density. Compared to the bimetallic systems, such as the Fe-Ni system, this Co-PH is a single-metal catalyst, thus the promoted OER activity results from the elevating population of Co^{4+} sites. Both experiments and DFT calculations show that phen could facilitate the generation of Co^{4+} sites. (2) stabilized Co^{4+} sites. Usually, high-valence metal species like Co^{4+} are soluble. Here, the phen could stabilize Co^{4+} on the electrode. Besides, the phen can make CoOOH amorphous and promote the charge transfer of CoO_2 motifs. (3) polaron effect. The binding energy of oxygenated intermediates on the Co^{4+} with non-bonding phen ligands is lower. Higher content of Co^{4+} could easier overcome the rate-determining step ($\text{OH}^* \rightarrow \text{O}^*$) for OER, where the phen-induced polaron with one Co^{3+} is in the region of enhanced content of Co^{4+} .
- Compared to the interesting reference, we agree with the referee that there is something in common, but we would like to emphasize the non-covalent interaction between oxide and ligand, which has been long ignored.
- We thank the referee for making us aware of some recent works. We have now considered these very interesting publications and referred to them where are suitable.

We have added (references 22 and 23) and referred to them on page 3 in the *main text*:

“To favor the formation of high-valence metal sites, heteroatom Fe^{13,14}, Cu¹⁵, Co¹⁶, Ni¹⁷, Au^{18,19}, or Ag²⁰ have been incorporated into the host metal oxides to lower the thermodynamic barrier^{3,8,21-23}.”

Added (reference 22) and referred to on page 7 in the *main text*:

“Importantly, pc-CoOOH presents ~1.8 times higher absorbance intensity for Co^{4+} (band at 730 nm) relative to pf-CoOOH at 1.75 V_{RHE} (Fig. 2e), indicating an easier charge accumulation on Co sites²², thus a much higher population of Co^{4+} .”

And added (reference 23) and referred to on page 15 in the *main text*:

“Thus, OER starts from the deprotonation of the surface OH^ , as has been proposed in the previous reports^{16,23,69}.”*

- I think the article would benefit from some insight into why this particular ligand was chosen? Why does the coordination of the Co site with this specific ligand result in it being more oxidized? It is unclear to me how and why the authors choose this ligand? What is the impact of this ligand on the binding energetics of oxygenated intermediates which are key to describe OER activity?

Responses 6:

- Thanks very much for these interesting questions. We have tried a series of ligands in this work (Fig. R18), and all of them improve the OER activity at different levels. However, simulating and calculating all these ligands are quite time-consuming to prove the non-covalent interaction. We thus did not show all of them in this work. To demonstrate an ideal model, we choose the phen (superior to the other ligands) as an example

to demonstrate the non-covalent interaction.

Fig. R18. The j - V curves of CoOOH and ligand-treated CoOOH. The ligand-treated CoOOH represents that the pristine CoOOH pre-catalysts were treated at 1.7 V_{RHE} for 10 h in ligand-containing 1.0 M NaOH electrolytes. The structure inset in the figures indicates the corresponding ligand. The grey j - V curves in all figures indicate the pristine CoOOH.

- Regarding the reasons why phen could promote Co oxidation, two points are important according to the discussions on pages 6-8 in the *main text*. (1) As shown in Supplementary Fig. 5 on page S6 in the *supplementary information*, the phen could promote the amorphization of CoOOH during the OER process, thus more CoO₂ fragments conjugated with phen (N in phen rich with electron); theoretically, phen decreases the overall formation energy of CoO₂. (2) The phen embedment increases the density of states near the Fermi level and thereby improves the charge transfer kinetics (Fig. 3c on page 8 in the *main text* and Supplementary Fig. 16 on page S19 in the *supplementary information*).
- Concerning the impact of phen on the binding energetics of oxygenated intermediates, we have supplemented further DFT calculations (Fig. R19). Benefiting from the interaction between phen and CoO₂, the reaction-free energy barrier of OH* oxidation to O* was reduced from 1.98 eV to 1.63 eV. This lowered energy barrier could favor the formation of O* (rate-limiting step).

Fig. R19. The reaction free-energy diagrams for OER on CoO₂ and Co-PH at 0 V_{RHE}. Insets are the corresponding enlarged models.

- The supplementary DFT calculations are shown in Fig. 7 on page 14 in the *main text*, and in Supplementary Fig. 52-65 on pages S60-S73 in the *supplementary information*. The corresponding discussion has been added on pages 14-15 in the *main text* as follows:

“Understanding improved activity through modeling. To better understand the role of phen, we further studied OER activity through DFT calculations. We first evaluated the steady-state configurations of CoO₂ (10-10) and Co-PH (10-10) surfaces (Fig. 7a). The calculated surface free energy diagrams of CoO₂ and Co-PH indicate that, under OER conditions, coordinatively unsaturated surface O sites are saturated with H_{ad} by forming bridge OH species, and coordinatively unsaturated Co sites are saturated with OH_{ad} by forming atop OH* (Fig. 7b and Supplementary Fig. 52). Thus, OER starts from the deprotonation of the surface OH*, as has been proposed in the previous reports^{16,23,70}. Also, we found that OH* deprotonation to O* is the potential limiting step for both bridge OH* and atop OH* (Fig. 7d and Supplementary Fig. 53). For the deprotonation of bridge OH*, the overpotential is about 0.2–0.6 V lower than that of atop OH* (Supplementary Fig. 54). Thus, OH bridged dual Co-Co sites are reaction centers.*

For CoO₂, the overpotential is 0.75 V (Fig. 7d). The presence of polaron-like Co³⁺ induced by phen, however, can reduce the overpotential to 0.4–0.5 V, which is dependent on the relative distance of the reaction center to the location of Co³⁺ site (Fig. 7c-7d and Supplementary Figs. 55–65). The reaction center with the lowest overpotential of 0.4 V is the one with Co³⁺ forming the Co³⁺-Co⁴⁺ reaction center. This type of reaction center facilitates the potential-limiting deprotonation of OH to O* which is accompanied by the oxidation of Co³⁺ to Co⁴⁺, as observed by the change of Co magnetic moment from 0 to 1 μB (Supplementary Figs. 58–61). However, even if the Co³⁺ site is not part of the reaction center, i.e., forming Co⁴⁺-Co⁴⁺ type reaction centers, the overpotential only increases slightly (Supplementary Figs. 62–65). For example, for the cases of Co³⁺ with one atom away from the reaction center, the overpotential only increases 0.02–0.05 V due to the effect of polaron, i.e., Co³⁺ is still oxidized to Co⁴⁺ during the deprotonation of OH* to O* (Supplementary Figs. 58–61). The feature of polaron is more clearly illustrated in the cases where Co³⁺ is far away from the reaction center, e.g., on the opposite side of the slab model. For these cases, the overpotential is only up to 0.1 V higher (Supplementary Figs. 62–65). These results highlight the importance of non-covalent ligand-oxide interactions in reducing the OER overpotential by generating abundant Co⁴⁺, active polarons, and dynamic reaction centers.”*

- Does the electrolyte for the Co-PH data in Figure 6 contain phen? In other words, is the Co-PH catalyst stable in electrolyte that does not contain the phen ligand?

Responses 7:

- Thanks very much for these questions. The electrolyte for the Co-PH stability test in Fig. 6 contains phen. The electrolyte details have been supplemented in the figure caption of Fig. 6b on page 13 in the *main text*.

The stability test of as-prepared CoOOH was operated in 1.0 M NaOH. Co-PH was in-situ deposited on bare FTO and its stability test was operated in Co(phen)₂(OH)₂-containing 1.0 M NaOH.

- We carried out the stability test of Co-PH in 1.0 M NaOH without phen. For example, the Co-PH is pre-deposited for 10 h on bare FTO in Co(phen)₂(OH)₂-containing 1.0 M NaOH at 10 mA cm⁻². Then the electrolyte is changed to 1.0 M NaOH. As shown in Fig. R20, the Co-PH (after 10 h deposition) also shows good stability over 1000 h in NaOH. Without phen in the electrolyte, we propose that the phen in the catalyst film may be lost thus the OER activity is relatively lower relative to that with phen in the electrolyte.

Fig. R20. The stability tests of Co-PH in 1.0 M NaOH at 10 mA cm⁻² on planar FTO. The Co-PH was *in-situ* deposited at 10 mA cm⁻² for 10 h in Co(phen)₂(OH)₂-containing 1.0 M NaOH.

- It is unclear if the Co sites that interact with the phen ligands are still able to bind reaction intermediates of OER or not?

Responses 8:

- Thanks very much for this very interesting question. From the DFT calculations, we find that the presence of phen induces the so-called polaron to influence the Co sites nearby, as shown in Fig. R21, with selected eight different reaction Co sites being influenced. As a result, the OER activity can be improved. We also reasoned that the Co sites to bind reaction intermediates might be influenced depending on the geometry of phen with the Co sites and that the geometry of phen will re-arrange to accommodate the reaction intermediates. Thus, we will take the view that it should be a dynamic and structurally self-adaptive process.

Fig. R21. (a) The reaction free-energy diagram of different Co sites in Co-PH. (b) The top view of Co-PH models.

REVIEWER COMMENTS

Reviewer #1 (Remarks to the Author):

The authors have adequately responded to my enquiry of including an atomistic model of the OER pathway for their catalyst. While in principle, the paper could be published, I have three technical comments that I ask the authors to consider within a minor revision:

i) The authors discuss the OER landscape based on thermodynamic considerations. Therefore, they are not resolving the RLS (as pointed out in the figure and text), but rather the potential-determining step (PDS). Please note the difference between PDS and RDS (rate-determining step), see for instance *Journal of Solid State Electrochemistry* volume 17, pages 339–344 (2013) or *ACS Catal.* 2020, 10, 21, 12607–12617.

If the authors aim to resolve the RDS, they need to connect their thermodynamic picture by applying a more sophisticated descriptor, namely $G_{\max}(\eta)$, with the experimental Tafel slope.

ii) Please do not write “the overpotential is ...” in the text. Overpotential serves as the driving force but is not the actual result of the electrocatalytic reaction. To be precise, the corresponding descriptor is named “theoretical overpotential” or “thermodynamic overpotential”.

iii) The OER takes only place for potentials exceeding the equilibrium potential, that is, $U > 1.23$ V vs. RHE. Therefore, it is advisable to translate the free-energy landscape to potentials that correspond to the ones encountered under experimental conditions.

Wu et al have provided a revised version of their article. I would have been happy to revise my negative initial assessment (see below paragraph) but unfortunately, I do not see the additional experiments that could lead to a change in my initial assessment. In the following, I list my original comments on the initially submitted version and briefly comment on the response of the authors to these concerns. Major points are highlighted by bold letters

This reviewer is sympathizing with the broad experimental approach and sees some merits of the study. However, taking into account the below list of questions and deficits regarding both significance of the author's conclusions and central technical aspects, publication cannot be recommended.

In parts this investigation provides valuable and convincing experimental evidence, specifically regarding the increased stability of the catalyst material when operated in a phen-containing electrolyte. I understood that this stability enhancement is explained as resulting from non-covalently bound phen which enhances the propensity for Co^{4+} formation and thereby the efficiency for self-healing by redeposition of Co^{2+} ions. Arguing that phen is non-covalently bound to (or within) the catalyst materials likely has led to the title suggesting on 'non-covalent ligand-oxide interactions' that promote OER.

It is a weak point in the author's argumentation that neither the non-covalent binding mode nor the intactness of catalyst-internal phen is convincingly shown. Therefore and because of lacking investigation of further compounds aside from phen, the title (and thus the main message of the study) is not sufficiently well backed up by the findings of the study.

Some additional sentences have been added to the manuscript text. The issue itself, which is crucial for the significance of the study, has not been resolved. The needed experimental evidence is, unfortunately, still lacking.

(Moreover, 'non-covalent ligand-oxide interaction' is an unclear expression that does not describe the suggested situation well.)

This point has been clarified.

A second especially critical point is the suggested performance increase by phen. An overpotential of 216 mV per cm^2 is reported (at 10 mA/cm^2). However, this low overpotential is present only after about 2 weeks of catalyst operation. Within the first hours of catalyst operation, however, the overpotential is much higher, exceeding 500 mV, as visible in Fig. S44. In the context of a study on the phen influences, publication would require convincing elucidation of the reasons of the dramatic performance increase that requires >10 days for completion.

The reason for dramatic performance increase that requires >10 days for completion are still not clarified. This is a truly crucial issue and clarification would require extensive additional experimentation; these have not been approached.

The role of uncontrolled Fe contaminations needs to be examined and/or thoroughly excluded.

This point has been clarified.

In addition, it will be important to clarify for all reported data the respective status of the catalyst material regarding its long-term activation state.

Has not been convincingly clarified.

Further problematic points:

- Co-O distances of 1.5 Å and 2.5 Å are reported, based on the EXAFS analysis. The first value is unreasonably short (might be in order for Co=O, but massive formation of terminal oxo species is unlikely). And the second value is unreasonably long. Similarly, also the coordination numbers are not especially reasonable. Moreover, insufficient details on the EXAFS simulation are provided. Also for submission to another journal, the EXAFS analysis would need to be approached anew. Earlier studies by others may provide guidelines, e.g. Risch et al, Energy Env. Science, 2015. Also, it should not be overlooked that the EXAFS cannot discriminate between oxygen and nitrogen ligation.

The EXAFS analysis has been updated in the revised manuscript and thereby above concern largely has been addressed. The obtained coordination numbers are not longer highly unreasonable. However, the evidence for coordination numbers of ca. 3.8 and ca. 5.0 is not highly convincing, especially as several earlier studies on similar materials by others strongly suggest prevalence of hexa-coordinate Co.

- In Fig. 2, a shift or redox waves is concluded from the CV data in Fig. 3. The underlying (mathematical) rationale needs to be explained. Right now it appears as if more or less arbitrary number are written into the CV figure.

An explanatory sentence which is meant to address this comment has been added to the manuscript. However, the explanation I was asking for is still lacking. To me the provided numbers still appear as being more or less arbitrary values.

The shift in redox potentials is not visible in panels e and f, or is it? Minor points: What exactly is shown in Figure 3d and what does it mean?

Some explanation is provided in the response letter.

- In Figure 3 and 4, there are panels showing lines and the data is called a phase diagram. Some unexplained ΔG value is plotted versus the applied potential. Figure caption does not say what ΔG is plotted, but just say that a phase diagram is shown. In any event, these are NOT phase diagrams (where lines typically separate areas of different phases).

This point has been addressed. The authors *inter alia* have corrected the erroneous use of the term "phase diagram".

- Densities of states are shown in Fig. 3c, with two red lines and two grey lines. What exactly is shown and what do these curves mean (why are they shown)?

Has been addressed.

- It is not clear whether the shown X-ray data for the 1.7 V vs. RHE catalysts has been obtained by operando spectroscopy, that is, whether the data was collected during application of the electrochemical potential. This should be stated very clearly. If it were not an operando experiment, this would weaken the impact of the data significantly.

The authors clarify in the response letter that indeed the XAS experiment had not been operando experiments. Unfortunately, the lack of operando experiments strongly weakens the relevance of the XAS data.

- In the Supporting information, all figure captions need to contain clearly more information about the shown data.

Has been addressed, at least partially. (I do not want to be pedantic, but I still think that even more complete information would be good.)

- In the structures shown in Fig. S18, the Co ion coordinated to a single phen are assumed to be four-coordinate. Why is six-coordinate Co not considered?

Has been addressed.

- In Fig. S21, why should 16O-18O exchange optical spectra. Maybe the observed difference occurred just by chance (irreproducibility of, e.g., background absorption).

The question of reproducibility is addressed by providing additional data that suggest reproducibility. A reasonable physical argument how 16O-18O exchange might affect UV-vis spectra is still lacking.

- In Fig. S24: It is doubtful that the assumed non-covalent coordination mode would result in disappearance of the sharp phen lines of the phen IR spectrum. The formation of broad bands in Co-PH could be reconciled more easily with a structure where phen or phen fragments coordinated to Co ions.

A convincing response is not provided.

Reviewer #3 (Remarks to the Author):

The authors have addressed my original comments satisfactorily. In particular, they have corrected the errors of EXAFS fitting and have provided DFT energetics for the OER mechanism.

Dear Reviewers:

We highly appreciate your careful review and constructive comments which help us to significantly improve the quality of this manuscript (NCOMMS-22-17715A). Followed by the suggestions from the second round, we have carried out more experiments, including *in-situ* XAFS measurements at the indicated potentials, the fast catalyst deposition process at higher current densities and the NMR. We carefully revised the manuscript accordingly. We made corresponding corrections point by point as follows (corrections in the revised manuscript are marked in blue color).

Reviewer #1 (Remarks to the Author):

The authors have adequately responded to my enquiry of including an atomistic model of the OER pathway for their catalyst. While in principle, the paper could be published, I have three technical comments that I ask the authors to consider within a minor revision:

Responses 1:

- We highly appreciate the positive comment and previous professional suggestions from the reviewer. The comments are quite helpful, please find our response and revision as follows.

i) The authors discuss the OER landscape based on thermodynamic considerations. Therefore, they are not resolving the RLS (as pointed out in the figure and text), but rather the potential-determining step (PDS). Please note the difference between PDS and RDS (rate-determining step), see for instance Journal of Solid State Electrochemistry volume 17, pages 339–344 (2013) or ACS Catal. 2020, 10, 21, 12607–12617.

If the authors aim to resolve the RDS, they need to connect their thermodynamic picture by applying a more sophisticated descriptor, namely $G_{\max}(\eta)$, with the experimental Tafel slope.

Responses 2:

- Thanks very much for your careful review and pointing out this issue. We have carefully corrected the words “rate-determining step” to “potential-determining step” in the *main text* and *supplementary information*. The corresponding notes in the figures in the *main text* and *supplementary information* also are corrected.

- We greatly appreciate the reviewer for reminding us for distinguishing between the RDS and PDS.

ii) Please do not write “the overpotential is ...” in the text. Overpotential serves as the driving force but is not the actual result of the electrocatalytic reaction. To be precise, the corresponding descriptor is named “theoretical overpotential” or “thermodynamic overpotential”.

Responses 3:

- Thanks very much for this suggestion. We have corrected the words “overpotential” to “theoretical overpotential” and “thermodynamic overpotential” in the DFT calculations section in the *main text* and *supplementary information*.

- We greatly appreciate the reviewer for reminding us of this careless mistake.

iii) The OER takes only place for potentials exceeding the equilibrium potential, that is, $U > 1.23$ V vs. RHE. Therefore, it is advisable to translate the free-energy landscape to potentials that correspond to the ones encountered under experimental conditions.

Responses 4:

- Thanks very much for this insightful suggestion. We have changed the free-energy landscape into corresponding potentials in the DFT calculations section in the *main text* and *supplementary information*.

Reviewer #2 (Remarks to the Author):

Wu et al have provided a revised version of their article. I would have been happy to revise my negative initial assessment (see below paragraph) but unfortunately, I do not see the additional experiments that could lead to a change in my initial assessment. In the following, I list my original comments on the initially submitted version and briefly comment on the response of the authors to these concerns. Major points are highlighted by bold letters.

Responses 1:

- We thank the reviewer for the positive evaluation of the previously revised manuscript. We highly appreciate the suggestions to improve the quality of this paper; those comments are all valuable and helpful for revising and improving this paper. Please find the corresponding response and revision below.
- Particularly, we supplemented *in-situ* XAFS measurements at the indicated potentials in this revision, the fast catalyst deposition process at higher current densities as well as the NMR data to further confirm the intactness of phen.

This reviewer is sympathizing with the broad experimental approach and sees some merits of the study. However, taking into account the below list of questions and deficits regarding both significance of the author's conclusions and central technical aspects, publication cannot be recommended.

In parts this investigation provides valuable and convincing experimental evidence, specifically regarding the increased stability of the catalyst material when operated in a phen-containing electrolyte. I understood that this stability enhancement is explained as resulting from noncovalently bound phen which enhances the propensity for Co^{4+} formation and thereby the efficiency for self-healing by redeposition of Co^{2+} ions. Arguing that phen is non-covalently bound to (or within) the catalyst materials likely has led to the title suggesting on 'non-covalent ligand-oxide interactions' that promote OER.

It is a weak point in the author's argumentation that neither the non-covalent binding mode nor the intactness of catalyst-internal phen is convincingly shown. Therefore and because of lacking investigation of further compounds aside from phen, the title (and thus the main message of the study) is not sufficiently well backed up by the findings of the study.

Some additional sentences have been added to the manuscript text. The issue itself, which is crucial for the significance of the study, has not been resolved. The needed experimental evidence is, unfortunately, still lacking.

Responses 2:

- We are so grateful for your kind question. Regarding the evidence of non-covalent interaction, we have carried out many DFT calculations containing the stabilization energy, interaction distance, and ΔG -potential diagrams in the manuscript. The concept of non-covalent interactions, as an energy-based notion, has already been widely reported in the DFT calculations area. For instance, Müller-Dethlefs et al. (*Chem. Rev.*, 2000, 100, 143-167) concluded that the theoretical description of non-covalent interactions, containing stabilization energy, equilibrium distance, $T\Delta S^0$ entropy term, etc. Marković et al. (*Nat. Chem.*, 2009, 1, 466-472) reported the cation-based non-covalent interactions by the stabilization energies based on the DFT calculations. Sun et al. (*Nano Energy*, 2022, 102, 107654-107662) explored the role of non-covalent interactions between the electrode surface and hydrated cations located in the outer Helmholtz plane on the HER kinetics by theoretical analysis. Particularly, our DFT calculation shows that the phen even cannot coordinate with Co^{4+} (energetically impossible) as a complex molecular (Supplementary Fig. 24). Thus, in general, it is useful to distinguish the covalent and noncovalent interactions by the interaction distance and stabilization energy based on the DFT calculations.

Regarding the experimental evidence of non-covalent interactions, in the second round of the revised manuscript, we demonstrated the valence-dependent coordination between Co sites and phen ligands. The phen can only coordinate with Co^{2+} thus forming soluble Co^{2+} -phen with phen in the electrolytes. While the phen cannot interact with Co^{3+} . However, the experiments reveal the interplay between phen with CoO_2 (Co^{4+}) (Supplementary discussion to Supplementary Figs. 20-23). Both the experiments and DFT calculations are consistent and well support the conclusion.

- To further confirm the intactness of phen in Co-PH film, except for the supplemented FT-ICR-MS measurements in the previously revised manuscript (Fig. R1), we further carried out the ^1H and ^{13}C NMR spectrum of Co-PH films in this revision. As shown in Fig. R2a, compared with the free phen, the dissolved Co-PH exhibited resonance signals with the same chemical shift in ^1H NMR spectra. And no other impurities resonance signals were found. In addition, the existence of phen molecule also was confirmed in ^{13}C NMR spectra (Fig. R2b). Therefore, we could conclude that the phen in Co-PH films is intact. The supplementary NMR spectra are shown in Supplementary Fig. 50 in the *supplementary information* and the corresponding discussion has been provided on page 13 in the *main text*.

Fig. R1. The FT-ICR-MS spectra of dissolved Co-PH film. The Co-PH film was deposited at $1.7 V_{\text{RHE}}$ in $\text{Co}(\text{phen})_2(\text{OH})_2$ -containing 1.0 M NaOH for 10 h. For the measurements of FT-ICR-MS, typically, deposited Co-PH film was first soaked in the 10 wt% HNO_3 solutions over 10 h to dissolve. Excess NaOH solid was then added to this solution to adjust the solution pH, resulting in the formation of 1.0 M NaOH (pH 13.9). The obtained solution was used for the FT-ICR-MS characterization. The spectra were recorded using dual electrospray ionization (ESI) source in the positive ion mode.

Fig. R2. (a) The ^1H and (b) ^{13}C NMR spectra of dissolved Co-PH film. The Co-PH film was deposited at 1.7 V_{RHE} in $\text{Co}(\text{phen})_2(\text{OH})_2$ -containing 1.0 M NaOH for 10 h. For the measurements of NMR measurements, typically, deposited Co-PH film was first soaked in the 5 wt% HNO_3 solutions containing 95 wt% D_2O over 10 h to dissolve. The obtained solution was used for the NMR measurements. Phen powder was also dissolved in the 5 wt% HNO_3 solutions containing 95 wt% D_2O for NMR tests. The insets were the enlarged spectra of dissolved Co-PH film.

(Moreover, ‘non-covalent ligand-oxide interaction’ is an unclear expression that does not describe the suggested situation well.)

This point has been clarified.

Responses 3:

- We thank the reviewer again for this kind comment.

A second especially critical point is the suggested performance increase by phen. An overpotential of 216 mV per cm^2 is reported (at 10 mA/cm^2). However, this low overpotential is present only after about 2 weeks of catalyst operation. Within the first hours of catalyst operation, however, the overpotential is much higher, exceeding 500 mV, as visible in Fig. S44. In the context of a study on the phen influences, publication would require convincing elucidation of the reasons of the dramatic performance increase that requires >10 days for completion.

The reason for the dramatic performance increase that requires >10 days for completion are still not clarified. This is a truly crucial issue and clarification would require extensive additional experimentation; these have not been approached.

Responses 4:

- We appreciate the reviewer's comments. To address this concern, we have provided more experimental data in this revision and added more discussions regarding the stability tests.

To accelerate the catalyst deposition rate, we could apply higher current densities. The catalyst deposition rate depends on the current densities or the applied potentials, as shown in Supplementary Fig. 32-33 in the *supplementary information*. As shown in Fig. R3, the Co-PH was *in-situ* deposited on bare FTO at 50 mA cm⁻², 100 mA cm⁻², and 200 mA cm⁻² for 5 hours in Co(phen)₂(OH)₂-containing 1.0 M NaOH, respectively, and thereafter was operated at 10 mA cm⁻² in Co(phen)₂(OH)₂-containing 1.0 M NaOH. After the pre-deposition process at higher current densities, the Co-PH catalyst exhibits well activity with an overpotential of 247 mV, 231 mV, and 220 mV at 10 mA cm⁻², respectively. Thus, the overall period of the Co-PH catalyst deposition gradually self-optimizes to the best state and the deposition process can be greatly shortened within a few hours. The corresponding experimental data was supplemented in Supplementary Fig. 52 in the *supplementary information* and the corresponding discussion has been provided on page 14 in the *main text* as follows:

“The catalyst deposition and the subsequent self-optimizing process at 10 mA cm⁻² are relatively slow and gradually tend to the best state, and this process could be accelerated by increasing the current densities (Supplementary Fig. 52).”

Fig. R3. The stability tests of Co-PH on the FTO. The Co-PH was *in-situ* deposited on bare FTO at 50 mA cm⁻², 100 mA cm⁻², and 200 mA cm⁻² for 5 h and thereafter operated at 10 mA cm⁻² in Co(phen)₂(OH)₂-containing 1.0 M NaOH, respectively.

- Within the whole operation period, the self-optimizing to the best state is relatively slow at 10 mA cm^{-2} . Based on the Co-PH mass-OER activity relationship, the enhanced OER activity is highly relevant to the mass loading. Finally, its OER activity tends to saturate within a long period of time. This is different from the pre-loaded catalyst. To clearly show the operation condition, we have added some sentences on pages 13-14 and in the figure caption of Fig. 6 in the *main text*, respectively, as follows:

“To better understand the whole process and activity variation trend of Co-PH, we present both the catalyst deposition process and the subsequent OER stability test in one curve in Fig. 6b.”

“The stability test of the as-prepared CoOOH was operated in 1.0 M NaOH. Co-PH was in-situ deposited on bare FTO and its stability test was operated in Co(phen)₂(OH)₂-containing 1.0 M NaOH.”

The role of uncontrolled Fe contaminations needs to be examined and/or thoroughly excluded.

This point has been clarified.

Responses 5:

- We are grateful again for this comment.

In addition, it will be important to clarify for all reported data the respective status of the catalyst material regarding its long-term activation state.

Has not been convincingly clarified.

Responses 6:

- Thanks very much for this suggestion. We have supplemented more details about the catalyst operation condition. All revised texts are marked in blue color. Such as:

“Co-PH was freshly deposited in Co(phen)₂(OH)₂-containing 1.0 M NaOH for 10 min at 1.7 V_{RHE}.” (Page 13 in the main text)

“The stability test of the as-prepared CoOOH was operated in 1.0 M NaOH. Co-PH was in-situ deposited on bare FTO and its stability test was operated in Co(phen)₂(OH)₂-containing 1.0 M NaOH.” (Page 13 in the main text)

“The overpotential of Co-PH at 10 mA cm^{-2} was obtained from the stability test in Fig. 6b.” (Page 13 in the main text)

“To better understand the whole process and activity variation trend of Co-PH, we present both the catalyst deposition process and the subsequent OER stability test in one curve in Fig. 6b.” (Page 13 in the main text)

“The catalyst deposition and the subsequent self-optimizing process at 10 mA cm^{-2} are relatively slow and gradually tend to the best state, and this process could be accelerated by increasing the current densities (Supplementary Fig. 52)” (Page 14 in the main text)

“The online GC was carried out in a gas-tight H-cell with a typical three-electrode system, and the content of O₂ during the in-situ deposition process of Co-PH on FTO was obtained in Co(phen)₂(OH)₂-containing 1.0 M NaOH.” (Page S59 in supplementary information)

Further problematic points:

- Co-O distances of 1.5 Å and 2.5 Å are reported, based on the EXAFS analysis. The first value is unreasonably short (might be in order for Co=O, but massive formation of terminal oxo species is unlikely). And the second value is unreasonably long. Similarly, also the coordination numbers are not especially reasonable. Moreover, insufficient details on the EXAFS simulation are provided. Also for submission to another journal, the EXAFS analysis would need to be approached anew. Earlier studies by others may provide guidelines, e.g. Risch et al, Energy Env. Science, 2015. Also, it should not be overlooked that the EXAFS cannot discriminate between oxygen and nitrogen ligation.

The EXAFS analysis has been updated in the revised manuscript and thereby above concern largely has been addressed. The obtained coordination numbers are not longer highly unreasonable. However, the evidence for coordination numbers of ca. 3.8 and ca. 5.0 is not highly convincing, especially as several earlier studies on similar materials by others strongly suggest prevalence of hexa-coordinate Co.

Responses 7:

- Thanks very much for this comment. Regarding the coordination numbers of Co-O and Co-Co in the previous manuscript, we found that similar coordination numbers of Co-O and Co-Co were reported previously, such as *Nat. Commun.*, 2020, 11, 2522-2531, *Nat. Energy*, 2022, 7, 765-773.
- As suggested by the reviewer in the comments, the *in-situ* XAFS experiments were supplemented and could provide stronger evidence to support. Thus, we have further implemented the *in-situ* XAFS measurements at the indicated potentials. And the previous EXAFS fitting results have been replaced in this revision. The corrected coordination numbers of Co-O and Co-Co are corrected to 6.05 at 1.7 V and 4.99 at 1.7 V (Fig. R4), respectively, which are consistent with the reviewer's suggestion. So now the supplemented *in-situ* XAFS measurements can support the conclusion (references in Supplementary Fig. 9). This *in-situ* XAFS demonstrated similar trend with the previous quasi-in situ tests (data collected when the applied potential was just removed), but the differences in Co chemical valences are larger. The images of the *in-situ* testing setup has been added in the supplementary information.

Fig. R4. Fitting curves of Co K-edge EXAFS in R spaces for pf-CoOOH at (a) OCP and (b) at 1.7 V_{RHE}. Fitting curves of Co K-edge EXAFS in R spaces for pc-CoOOH at (c) OCP and (d) at 1.7 V_{RHE}. The images of the *in-situ* XAFS equipment and the electrochemical cell (right, now in Supplementary Fig. 8e and f).

- All EXAFS analysis has been updated, please find the details in Supplementary Figs. 10-11, and Supplementary Table 1 on pages 11-13 in the *supplementary information*. The corresponding discussion was added to page 6 in the *main text* as follows:

“In addition, relative to pf-CoOOH, the Co-O coordination number of pc-CoOOH increased from 5.11 at OCP to 6.05 at 1.7 V_{RHE}”⁵⁰ (Fig. 2b, Supplementary Figs. 10-11 and Supplementary Table 1)”

- In Fig. 2, a shift or redox waves is concluded from the CV data in Fig. 3. The underlying (mathematical) rationale needs to be explained. Right now it appears as if more or less arbitrary number are written into the CV figure.

An explanatory sentence which is meant to address this comment has been added to the manuscript. However, the explanation I was asking for is still lacking. To me the provided numbers still appear as being more or less arbitrary values.

Responses 8:

- Thanks very much for this concern. As the CV data were shown in Fig. R5a, we got the potential values of Co²⁺/Co³⁺ oxidation peak in pf-CoOOH and pc-CoOOH, i.e., 1.13 V and 1.09 V, respectively (Refer to *J. Am. Chem. Soc.*, 2020, 142, 12087-12095; *Energy Environ. Sci.*, 2022,15, 206-214). Following these works, as shown in Fig. R5, we also wanted to obtain the oxidation peak potential of Co³⁺ to Co⁴⁺.

Fig. R5. (a) First cyclic CVs measured for pure Co₃O₄ and V₀-Co₃O₄ in 1 M KOH (*J. Am. Chem. Soc.*, 2020, 142, 12087-12095). (b) Cyclic voltammograms of CoFeO_xH_y samples containing 0%, 15%, and 30% Fe were obtained at a scan rate of 100 mV s⁻¹ in 0.1 M Fe-free KOH electrolyte. At around 1.2 V, the pairs of redox events are attributed to the Co²⁺/Co³⁺ redox couple. The second redox events in 1.4 V to 1.55 V are assigned to the Co³⁺/Co⁴⁺ redox couple. The onset potentials for the oxidation to Co⁴⁺ depend on the Fe content. The more Fe is added, the earlier the Co³⁺ is oxidized (*Energy Environ. Sci.*, 2022,15, 206-214).

However, the oxidation peak of Co³⁺/Co⁴⁺ was overlapped seriously by the OER curves and could hardly be distinguished. Therefore, we compared the onset potential for the oxidation of Co³⁺ to Co⁴⁺. The corresponding onset potentials were obtained by the enlarged insets (partial range of CV curves) in Fig. R6a (Refer to *Energy Environ. Sci.*, 2022,15, 206-214). In order to amplify the onset oxidation curves of Co³⁺ to Co⁴⁺, we demonstrated the log *j*-potential curves in Fig. R6b. From Fig. R6, we could clearly observe the onset potential for the oxidation of Co³⁺ to Co⁴⁺, i.e., 1.31 V for pf-CoOOH and 1.26 V for pc-CoOOH.

Fig. R6. (a) The redox couples of pf-CoOOH and pc-CoOOH in 1.0 M NaOH, the enlarged insets represent the onset oxidation potential from Co^{3+} to Co^{4+} . (b) The log j -potential curves of pf-CoOOH and pc-CoOOH in 1.0 M NaOH.

The shift in redox potentials is not visible in panels e and f, or is it? Minor points: What exactly is shown in Figure 3d and what does it mean?

Some explanation is provided in the response letter.

Responses 9:

- We sincerely thank the reviewer again for this comment.

- In Figure 3 and 4, there are panels showing lines and the data is called a phase diagram. Some unexplained delta-G value is plotted versus the applied potential. Figure caption does not say what deltaG is plotted, but just say that a phase diagram is shown. In any event, these are NOT phase diagrams (where lines typically separate areas of different phases).

This point has been addressed. The authors inter alia have corrected the erroneous use of the term “phase diagram”.

Responses 10:

- We greatly thank the reviewer again for this comment.

- Densities of states are shown in Fig. 3c, with two red lines and two grey lines. What exactly is shown and what do these curves mean (why are they shown)?

Has been addressed.

Responses 11:

- We are grateful again for this comment.

- It is not clear whether the shown X-ray data for the 1.7 V vs. RHE catalysts has been obtained by operando spectroscopy, that is, whether the data was collected during application of the electrochemical potential. This should be stated very clearly. If it were not an operando experiment, this would weaken the impact of the data significantly.

The authors clarify in the response letter that indeed the XAS experiment had not been operando experiments. Unfortunately, the lack of operando experiments strongly weakens the relevance of the XAS data.

Responses 12:

- Thanks very much for this concern. We have supplemented in situ XAFS experiments at the indicated potentials in this revision. To obtain the in situ XAFS data, the carbon paper was used as a work substrate in XAFS measurements instead of previous conductive FTO glass. The in situ XAFS data was recorded by fluorescence mode in a typical three-electrode system. The new XAFS data has been updated in Figs. 2a-2b on page 7 in the *main text*, and supplemented in Supplementary Figs. 8 and 10-11 on pages S9 and S11-12 in the *supporting information*. We added the detailed experimental method to page 20 in the main text to make the experiment section clear as follows:

“In-situ X-ray Absorption Fine Structure (XAFS) measurements. All the XAFS data were collected at the BL11B beamline of the Shanghai Synchrotron Radiation Facility (SSRF). The beam current of the storage ring was 220 mA in a top-up mode and the incident photons were monochromatized by a Si (111) double-crystal monochromator, with an energy resolution $\Delta E/E \sim 2 \times 10^{-4}$. The spot size at the sample was $\sim 200 \mu\text{m} \times 250 \mu\text{m}$ ($H \times V$). The position of the absorption edge (E_0) was calibrated by using Co foil. To preclude the influence of the FTO and obtain the in-situ XAFS data, the carbon paper was used as a substrate. All XAFS spectra were collected in fluorescence mode. To monitor the changes of the catalyst during the OER process, different anodic potentials were applied to the catalyst in 1.0 M NaOH in a typical three-electrode system. After the corresponding current density of the catalyst reaches a steady state, the XAFS data can be collected at the indicated potentials. All XAFS data were analyzed by the ATHENA and ARTEMIS modules implemented in the IFEFFIT software package⁷⁵.”

- In the Supporting information, all figure captions need to contain clearly more information about the shown data.

Has been addressed, at least partially. (I do not want to be pedantic, but I still think that even more complete information would be good.)

Responses 13:

- Thanks very much for this comment. We double-checked again all figure captions and supplemented more details. All revised texts are marked in blue color. We sincerely hope that the details provided will address the reviewer's concerns.

- In the structures shown in Fig. S18, the Co ion coordinated to a single phen are assumed to be fourcoordinate. Why is six-coordinate Co not considered?

Has been addressed.

Responses 14:

- We greatly thank the reviewer again for this comment.

- In Fig. S21, why should 16O-18O exchange optical spectra. Maybe the observed difference occurred just by chance (irreproducibility of, e.g., background absorption).

The question of reproducibility is addressed by providing additional data that suggest reproducibility. A reasonable physical argument how 16O-18O exchange might affect UV-Vis spectra is still lacking.

Responses 15:

- Thanks very much for this concern. The isotope effects on the UV-Vis spectrum have been reported in previous works. Morisawa et al. (*Chem. Phys. Lett.*, 2009, 476, 205-208) found remarkable isotope effects in the FUV spectra of CH₃OH, CH₃OD, CD₃OH, and CD₃OD, as shown in Fig. R7a. The corresponding peak

position showed a significant blue shift upon the deuterations. Jana et al. (*Sci. J. Anal. Chem.*, 2015, 3, 109-114) also reported the heavier ^{18}O isotope can lead to the blue shift of C=O stretching frequency (Fig. R7b). These isotope effects might originate from the difference in vibrational zero-point energy, as shown in Fig. R7c, the peak of D_2O lies at higher energy than that of H_2O (*Rev. Sci. Instrum.*, 2007, 78, 103107-103112). Thus, the UV-Vis spectrum can be used to help the assignments of the absorption spectra of molecules with different isotopes.

For our UV-Vis spectra of $\text{Co}(\text{phen})_2(^{16}\text{OH})_2$, $\text{Co}(\text{phen})_2(^{18}\text{OH})_2$, and $\text{Co}(\text{phen})_2(\text{OD})_2$ in Supplementary Fig. 27 in the *supplementary information*, the peak at ~ 506 nm shows a similar shift due to this isotope effect.

In order to make the interpretation more reasonable, we added the above reference and modified the corresponding discussion of Supplementary Fig. 27 in the *supplementary information* as follows:

“The peak at ~ 506 nm shows a slight shift due to the isotope effect⁹⁻¹¹, indicating the water-associated coordination.”

Fig. R7. (a) ATR-FUV spectra in the region from 145 to 200 nm of CH_3OH (solid), CH_3OD (break), CD_3OH (dot break) and CD_3OD (short break) in the pure liquid state (*Chem. Phys. Lett.*, 2009, 476, 205-208). (b) UV-Vis spectra of control and treated samples of benzophenone (*Sci. J. Anal. Chem.*, 2015, 3, 109-114). (c) ATR-FUV spectra of light and heavy water in the 140-200 nm region obtained with the spectrometer (*Rev. Sci. Instrum.*, 2007, 78, 103107-103112).

- In Fig. S24: It is doubtful that the assumed non-covalent coordination mode would result in disappearance of the sharp phen lines of the phen IR spectrum. The formation of broad bands in Co-PH could be reconciled more easily with a structure where phen or phen fragments coordinated to Co ions.

A convincing response is not provided.

Responses 16:

- Thanks very much for this comment. To clarify the phenomenon, we carried out more experiments as follows:

1) We first repeated the *ex-situ* ATR FTIR measurements of Co-PH catalyst on FTO. From Fig. R8, we found similar vibration bands of phen in the Co-PH catalyst. These vibration bands further determined the presence

of phen in Co-PH film, as presented in the previous revision.

Fig. R8. (a) The *ex-situ* ATR-FTIR spectra of Co-PH film on FTO and phenanthroline powder in the previous manuscript. (b) The latest *ex-situ* ATR-FTIR spectra of Co-PH film on FTO and phenanthroline powder. The ATR-FTIR spectrum of phenanthroline powder is measured directly by pressing.

2) Regarding the broad bands of ATR-FTIR observed in Co-PH film on FTO, we deduce that the transparent FTO substrate and thickness of samples would significantly affect the vibration bands' strength. Thus, we directly prepared the phenanthroline film on FTO with different thicknesses by spin-coating method. From Fig. R9, compared with the phenanthroline powder, we found that the phenanthroline film on the FTO substrate also exhibited broad vibration bands. In addition, stronger and more sharp vibration peaks were observed in the thicker phenanthroline. Thus, the thickness or the amount (dispersion) of the phenanthroline on the FTO is crucial.

Fig. R9. The *ex-situ* ATR-FTIR spectra of phenanthroline film on FTO with different thicknesses and phenanthroline powder. The phenanthroline film on FTO was prepared by the spin-coating method. Typically, the 14 mM phenanthroline was dissolved in DI water, and 1 mL solution was spin-coated each time onto FTO at 500 rpm for 10 seconds thrice followed by at 3000 rpm once for one minute. The phenanthroline film with different thicknesses was prepared by repeating the whole spin-coating process, i.e., 5 times (blue curves) and 10 times (red curves). The ATR-FTIR spectrum of phenanthroline powder was measured directly by

pressing.

3) We also considered the ATR-FTIR spectra of well-defined $\text{Co(phen)}_3\text{Cl}_2$ with covalent interactions in Fig. R10. The obtained spectra exhibited the same trend, i.e., the covalent $\text{Co(phen)}_3\text{Cl}_2$ film on the FTO substrate displayed broad vibration bands, while the sharp peaks were significantly observed in the corresponding powder sample. That is, it is difficult to conclude that sharp or broad vibration bands are associated with non-covalent interactions or covalent interactions.

Fig. R10. (a) The *ex-situ* ATR-FTIR spectra of $\text{Co(phen)}_3\text{Cl}_2$ film on FTO and phenanthroline powder. (b) The *ex-situ* ATR-FTIR spectra of $\text{Co(phen)}_3\text{Cl}_2$ powder and phenanthroline powder. The $\text{Co(phen)}_3\text{Cl}_2$ film on FTO was prepared by the spin-coating method. Typically, 1 mL $\text{Co(phen)}_3\text{Cl}_2$ solution was spin-coated each time onto FTO at 500 rpm for 10 seconds thrice followed by at 3000 rpm once for one minute, this spin-coating process was repeated 5 times. The ATR-FTIR spectrum of powder was measured directly by pressing.

Reviewer #3 (Remarks to the Author):

The authors have addressed my original comments satisfactorily. In particular, they have corrected the errors of EXAFS fitting and have provided DFT energetics for the OER mechanism.

Responses 1:

- We sincerely appreciate the positive comment and professional suggestions from the reviewer.

REVIEWERS' COMMENTS

Reviewer #2 (Remarks to the Author):

The now provided information will allow the reader to judge by herself/himself the significance of the conclusions. In this sense, the revision has addressed my concerns sufficiently well.